# TBC9, an essential TBC-domain protein, regulates early vesicular transport and IMC formation in *Toxoplasma gondii*
Ming Sun ⓘ , Tao Tang, Kai He & Shaojun Long ⓘ ✉

Apicomplexan parasites harbor a complex endomembrane system as well as unique secretory organelles. These complex cellular structures require an elaborate vesicle trafficking system, which includes Rab GTPases and their regulators, to assure the biogenesis and secretory of the organelles. Here we exploit the model apicomplexan organism *Toxoplasma gondii* that encodes a family of Rab GTPase Activating Proteins, TBC (Tre-2/Bub2/Cdc16) domain-containing proteins. Functional profiling of these proteins in tachyzoites reveals that TBC9 is the only essential regulator, which is localized to the endoplasmic reticulum (ER) in *T. gondii* strains. Detailed analyses demonstrate that TBC9 is required for normal distribution of proteins targeting to the ER, and the Golgi apparatus in the parasite, as well as for the normal formation of daughter inner membrane complexes (IMCs). Pull-down assays show a strong protein interaction between TBC9 and specific Rab GTPases (Rab11A, Rab11B, and Rab2), supporting the role of TBC9 in daughter IMC formation and early vesicular transport. Thus, this study identifies the only essential TBC domain-containing protein TBC9 that regulates early vesicular transport and IMC formation in *T. gondii* and potentially in closely related protists.

*Toxoplasma gondii* is a highly prevalent parasite worldwide. It has a facultative life cycle and can infect almost all warm-blooded animals, including humans[1]. The life cycle of *T. gondii* is complex, involving two infectious stages in intermediate hosts: the tachyzoites that is responsible for acute infection, and the bradyzoites (in tissue cysts) that cause a chronic infection[2]. Although most of *T. gondii* infections are asymptomatic, latent infections in the brain and muscle tissues can lead to severe symptoms in immunocompromised patients[3]. Congenital infections can also result in serious consequences for the fetus, such as abortion and deformities[4]. More importantly, *T. gondii* can serve as a model organism for studying members of the Apicomplexa phylum, due to its ease of in vitro culturing and genetic manipulation. The Apicomplexa phylum consists of a large class of intracellular parasites comprising over 5000 species, including *Plasmodium spp.*, *T. gondii,* and *Cryptosporidium spp.*[5,6]. These parasites have evolved unique functional organelles, such as the micronemes, the rhoptries, the dense granules, and the apicoplasts, as well as other specialized cell structures (i.e., the inner membrane complex, the micropore and apical complex)[7–9]. These adaptations allow many biological processes of eukaryotic cells, particularly protein sorting, secretory processes as well as endocytic processes, to be specialized in parasites[7,10]. The secretory pathway of *T. gondii* exhibits a high level of polarity[10]. Membrane vesicles originate from the endoplasmic reticulum (ER) and are directed towards the closely juxtaposed Golgi stack[10].

Subsequently, they traverse through the Golgi apparatus, reaching the endosomal compartment while accurately targeting their respective destinations, namely the micronemes, the rhoptries, and the dense granules[10,11]. This vesicle-mediated transport of substances plays a crucial role in maintaining the normal functioning of parasite life activities. To ensure the accuracy of vesicle-based transport, it requires the synergistic action of various transport factors.

As the prominent subgroup of the small GTPase family, Rab GTPases are evolutionarily conserved GTPases found in eukaryotic cells, and they play a central role in regulating membrane traffic[12]. Similar to other small GTPases, Rab GTPases require the positive regulation of guanine nucleotide exchange factors (GEFs) and the negative regulation of GTPase-activating proteins (GAPs)[12,13]. Human cells express over 60 distinct Rab GTPases, while the *Saccharomyces cerevisiae* genome encodes 11 Rab GTPases[14–16]. Various Rab GTPases exert control over processes, which include vesicle budding, cargo sorting, transport, tethering, and fusion with targeted compartments by recruiting diverse effectors, such as sorting adaptors, tethering factors, kinases, phosphatases, and motor proteins[17,18]. In *T. gondii*, the genome encodes 15 Rab GTPases, of which 12 are expressed during the tachyzoite stage and primarily localized within the early and late secretory system[19]. Notably, Rab5A and Rab5C serve as crucial regulators in the microneme and rhoptry trafficking, while Rab6 governs protein transport

National Key Laboratory of Veterinary Public Health Safety and College of Veterinary Medicine, China Agricultural University, Beijing 100193, China.
✉e-mail: LongS2018@163.com

between post-Golgi dense granule organelles and the Golgi[19,20]. Rab11B is necessary for the inner membrane complex (IMC) biogenesis during daughter parasite formation, and vesicle transport mediated by Rab11A is vital for the IMC maturation and delivery of a new plasma membrane to daughter cells, thereby completing cell division[21,22]. Rab1B was recently found to be associated with the endocytic trafficking as well as vesicle sorting to the rhoptry bulb, the process of which was regulated by protein prenylation[23,24]. Moreover, vacuolar protein sorting 9 (*Tg*Vps9) has been identified as the GEF of Rab5a in *T. gondii*, playing a crucial role in sorting proteins destined for secretory organelles[25].

Rab GTPases have been analyzed due to the critical roles in biogenesis of the specialized structures, while the RabGAPs remain obscure. RabGAPs play a crucial role in stimulating GTP hydrolysis by releasing inorganic phosphate, thereby inducing a conformational change in Rab GTPases from an active GTP-bound state to an inactive GDP-bound state[26,27]. Many RabGAPs share a Tre-2/Bub2/Cdc16 (TBC) domain architecture, consisting of approximately 200 amino acids, which was originally identified as a common conserved domain in the tre-2 oncogene and the yeast regulators of mitosis, BUB2, and cdc16[28,29]. Proteins containing the TBC domain are widely conserved in eukaryotic organisms and are involved in various cellular processes such as cancer, cell signaling, and other intracellular activities in mammalian cells[13,30,31]. A subset of TBC proteins function as GAPs for Rab GTPases, known as TBC/RABGAPs[13]. TBC/RABGAPs accelerate GTP hydrolysis through a dual-finger mechanism, wherein an exposed arginine residue plays a critical role in GAP activity both in vitro and in vivo, and the other finger residue, known as the glutamine finger, substitutes for the glutamine in the DxxGQ motif of the GTPase[27,31,32]. These characteristics are highly conserved in all yeast TBC domains as well as in at least 70% of mammalian TBC domains[31]. Besides TBC domains, TBC/RABGAPs often harbor several other domains, which may contribute to the functional diversity of the TBC/RABGAPs family[33,34]. The first reported TBC/RabGAP is Gyp6, which functions as the GAP for ypt6 in *Saccharomyces cerevisiae*[26]. In mammalian cells, RabGAP-5 has been identified as the specific GAP for Rab5, governing endosomal trafficking by restricting the activation of Rab5[35]. In addition, OATL1 has been identified as the Rab33B-GAP and its involvement in regulating the auto-phagosomal maturation has been described[36]. Moreover, the GAP (TBC1D9B) for Rab11a has also been identified, and its regulatory role in basolateral-to-apical transcytosis in polarized MDCK cells has been elucidated[37].

In contrast to the extensively studied GAPs in yeast and mammals, their presence and function in protozoa basically remain unexplored. In this study, we conducted a systematic analysis of TBC domain-containing proteins in the model apicomplexan *T. gondii*. Among the 17 TBC domain-containing proteins found in the parasite genome, we identified and explored the only essential one TBC9. We demonstrated that TBC9 is mainly localized to the ER, and its depletion by the auxin-inducible degron system led to a fatal growth arrest, disruption of IMC formation, and destruction of the early protein sorting pathway-associated organelles, such as the ER and the Golgi. Through further analysis, it was found that Rab2 directly interacts with TBC9, providing evidence for its potential role in coordinating cargo transport within the early protein sorting pathway between the ER and the Golgi. Additionally, the interaction between RAB11A, RAB11B, and TBC9 supports the involvement of TBC9 in regulating vesicular trafficking during the formation and maturation processes of the daughter IMC. These findings contribute to understanding of the intricate cellular processes in *T. gondii* and potentially in closely related protists.

## Results
### Bioinformatic analysis of TBC domain-containing proteins in *T. gondii*
We identified 17 TBC proteins (TBC1-10, 12–18) in the genome of *T. gondii* by searching ToxoDB using the InterPro ID IPR035969 Rab-GAP-TBC domain superfamily (Fig. 1 and Supplementary Table 1). In the InterPro analysis, we observed that two candidates contain a Sec7 domain, one with a

TLD domain, five with transmembrane domains, and one with a protein kinase domain (Fig. 1). In addition, nearly half of the TBC proteins possess a coiled-coil (CC) domain (Fig. 1). The Sec7 domain is associated with the guanine nucleotide exchange factor (GEF) known to activate Arf GTPase[38]. Proteins containing both TBC and Sec7 domains are termed TBC-Sec7 (TBS) proteins, representing a subfamily of ArfGEFs[39]. In here, TBC16 and TBC17, two TBS proteins, potentially possess both GEF and GAP functions (Fig. 1).

We performed further bioinformatic analyses, from which we found that 12 of the TBC domain-containing proteins contain the "arginine finger" (Arg in the IxxDxxR motif) and "glutamine finger" (Gln in the YxQ motif) (Supplementary Table 1 and Supplementary Fig. 1). These fingers are critical for catalysis of GTP hydrolysis of Rab GTPases[31], indicating that these candidates are highly likely to be active GAP proteins in the parasite. These analyses showed that three of these TBC proteins contain only one active site, while the two remaining TBC proteins lack the dual-finger mechanism and may function through alternative active sites (Supplementary Table 1 and Supplementary Fig. 1). Notably, TGGT1_239830 was annotated as a TBC domain-containing protein in the ToxoDB, yet a TBC domain was unable to be identified by the InterPro, NCBI, OrthoMCL DB and UniProt. It was therefore not included in this study as a TBC domain-containing protein.

### Analyses of localization and essentiality of the TBC proteins in *T. gondii*
To determine the intracellular localization of TBC proteins in *T. gondii*, we performed endogenous tagging of the genes at the 3'-termini (Supplementary Fig. 2a) or 5'-termini (Supplementary Fig. 2b) of the coding sequences by adding the genes with a 6TY tag using a CRISPR/Cas9 approach, as described in our previous study and in the diagram[40–42]. The diagnostic PCR confirmed the accurate tagging of the Ty epitope at the end of the genes (Supplementary Fig. 2c and Supplementary Data 1). Immunofluorescence assays (IFA) showed that the TBC1, 5, 6, 9 and 14 were primarily located in the cytoplasmic region and the endoplasmic reticulum (the ER) (Fig. 2a). Co-localization studies with a ER marker BIP[43] showed at least a partial co-localization with the ER was observed for these TBC proteins (Fig. 2a). The statistics analysis of Pearson correlation coefficient (PCC) over the fluorescent region and the zoom in the ER region provides evidence (Fig. 2b and Supplementary Fig. 2d). Furthermore, TBC2, 3, and 18 were localized to foci anterior to the nucleus, which were likely to be the Golgi, while TBC3 also exhibited a partial distribution in the cytoplasm (Fig. 2b). Co-localization analysis with the cis-Golgi marker GRASP[44] and the trans-Golgi network (the TGN) marker STX6[45] showed that TBC2, 3, and 18 signal was more closely associated with the TGN than the cis-Golgi (Fig. 2c). The statistics analysis of PCC over the fluorescent foci provided strong evidence supporting the co-localization of the proteins (TBC2, 3, and 18) with the TGN (Fig. 2d). In addition, we also observed that seven TBC proteins (TBC4, 8, 10, 12, 13, 16 and 17) displayed cytoplasmic localizations with punctate or vesicular staining (Fig. 2e). Intriguingly, TBC10 was demonstrated to have a partial localization in the residual body (Fig. 2e), while TBC15 exhibits localization at the daughter IMC (Fig. 2f). Finally, TBC7 was unable to be detected by IFA, although the diagnostic PCR confirmed the proper integration of the 6TY tag in the genes' termini (Supplementary Fig. 2c). This observation indicated a low expression level of the genes in the parasite, which is consistent with data derived from the ToxoDB.

The parasite fitness scores retrieved from a CRISPR library screening indicated that most of the genes are more likely to be non-essential (Fig. 3a)[6]. To generate the knockouts of the genes in the parasite, we employed a CRISPR/Cas9-mediated knockout strategy using either double or single sgRNAs (Supplementary Fig. 3a, b), as described in our previous study[46] and in the diagram. The diagnostic PCR was used to select and confirm the parasite lines of gene knockouts (Supplementary Fig. 3c). Ultimately, we successfully generated knockout lines for TBC1-8, 10, and 12–18, yet we were unable to delete the *TBC9* gene even with several attempts, suggesting

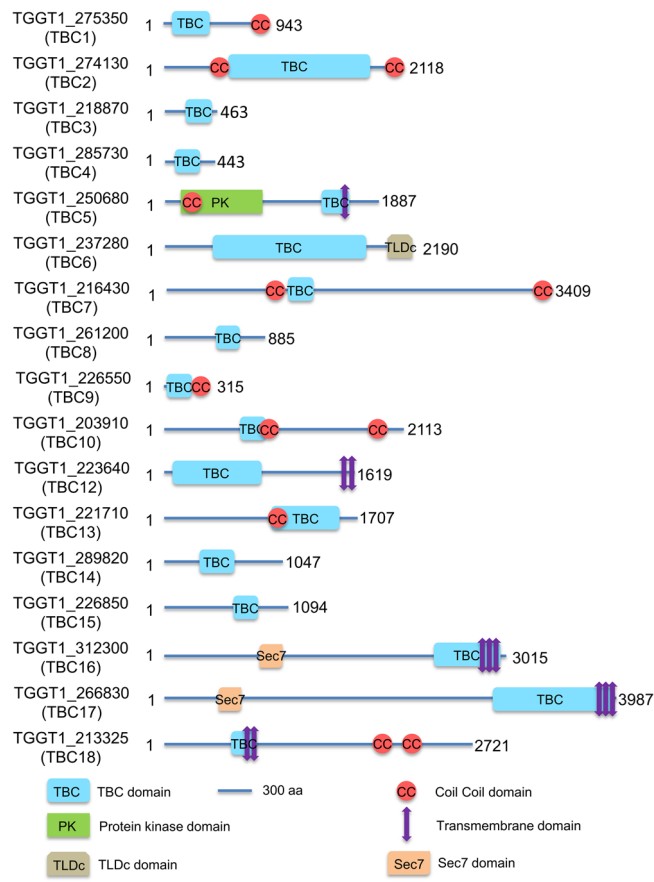

**Fig. 1 | TBC domain-containing proteins in *Toxoplasma gondii*.** The schematic diagram illustrates the domain structure and other features of the 17 TBC proteins as predicted by the InterPro (https://www.ebi.ac.uk/interpro/). TBC - Tre2-Bub2-Cdc16 domain; CC - coiled coil domain; PK - protein kinase domain; Transmembrane domain - region of a membrane-bound protein predicted to be embedded in the membrane; TLDc - a domain that includes two conserved domains: TBC and LysM; Sec7 - a conserved domain of GEF (guanine nucleotide exchange factors). A scale of 300 amino acids length was shown for the protein length displayed in the schematic.

that TBC9 is essential to the parasite (Fig. 3b). To assess the impact of these knockouts on parasite growth, we performed plaque formation assays on host cell monolayers over a 7-day period (Fig. 3b). The plaque quantification analysis showed that knockout of TBC1, 3, 5, 6, 15 and 17 significantly impaired the capability of parasite plaque formation by examining the plaque size and numbers (Fig. 3b–d). In addition, deletions of TBC2, 4, 12, 14, and 16 significantly reduced the plaque size in relative to the parental line (Fig. 3c). In conclusion, our analyses suggested that nearly half of the TBC proteins are localized in cellular structures associated with protein secretion or protein sorting pathways, and that part of these proteins are fitness conferring to the lytic cycle of the parasite in host cells.

## TBC9 is essential for parasite replication

Since TBC9 is refractory to CRISPR/Cas9 deletion in the parasite, we then focused on the detailed analysis of TBC9 on its function. Phylogenetic analyses showed that the orthologues of TBC9 can be identified in apicomplexan, amebozoan, and trypanosomatid genomes (Supplementary Fig. 4a), suggesting the high conservation of this protein in the major clades of parasites. This phylogenetic analysis delineated the established relationships among these taxa (Supplementary Fig. 4a). To facilitate the functional analysis of TBC9 in the parasite, we utilized the auxin-inducible degron (AID) system for a conditional knockdown of the protein in *T. gondii*, as described in our previous study[41,47,48]. The C-terminus of TBC9 was endogenously tagged with miniAID-3HA in the TIR1 parental line using a

CRISPR/Cas9 strategy (Fig. 4a). The TBC9-mAID fusion appeared to have a similar localization to the TBC9-Ty, which is an ER-localized protein (Figs. 2a, 4b). The mAID fusion line was then assessed by IFA and Western blot concerning the efficiency of protein degradation upon induction by indole-3-acetic acid (IAA). The IFA analysis for parasites induced for 24 h showed that the mAID fusion was degraded to an undetectable level (Fig. 4b). The parasites were then analyzed by Western blot using a series of IAA induction time points (1, 2, 3, 4, 6 h), which demonstrated an efficient degradation of the target protein TBC9-mAID fusion at the starting time point 1 hour (Fig. 4c). Subsequently, the mAID fusion line was examined on the plaque formation capability by growing the parasites on HFF cell monolayer for 7 days. Expectedly, the parasites grown in auxin exhibited undiscernible plaques on the monolayer (Fig. 4d), which is consistent with the CRISPR fitness scoring data. In contrast, the non-induced parasites and the parental line appeared to be normal on the plaque formation. The plaque formation is resulted from the lytic cycle of parasite invasion, replication and egress on host cell monolayer. We then examined the parasite replication by growing the parasites in auxin for 24 h and by analyzing the parasite numbers in single vacuoles. This analysis showed a sharp change of the parasite numbers in vacuoles for the mAID fusion parasites grown in auxin in comparison to the non-induced parasites (Fig. 4e). This result suggested that depletion of TBC9-mAID dramatically affected the parasite replication on host cells.

Compared to type I strains (GT1 and RH), type II strains (e.g., ME49) have lower virulence but are most commonly associated with toxoplasmosis in humans and animals[49,50]. To examine the role of TBC9 in a more prevalent strain, we generated a TBC9 knockdown line ME49Δ*ku80*Δ*hxgprt*/TIR1/TBC9-mAID-3HA, using the ME49Δ*ku80*Δ*hxgprt*/TIR1 line as the parental line[51]. The AID fusion in the ME49 parental line displayed similar ER localization to the RH strain (Supplementary Fig. 4b). After treatment with IAA for 24 h, TBC9 was efficiently degraded in the ME49 strain, resulting in the presence of only one parasite/vacuole, suggesting that depletion of TBC9 also halted parasite replication in the ME49 strain (Supplementary Fig. 4b). Plaque formation assay showed that ME49 parasites lost the ability to form plaques on HFF monolayer in the presence of auxin (Supplementary Fig. 4c), further supporting that the TBC9 depletion caused severe growth defects in the ME49 strain. Taken together, further experiments in the type II strain ME49 supported that TBC9 is essential in *T. gondii* lytic cycle.

## Depletion of the TBC9-mAID fusion resulted in defects of the organelles associated with protein sorting pathways

Since TBC9 is predicted to act on vesicle trafficking by regulating Rab GTPase activity, we analyzed various cellular structures using protein markers specific to these structures or known proteins involved in secretory trafficking. We observed that depletion of TBC9 in parasites, grown in auxin for 13 h after 16 h of growth in regular medium, resulted in the irregularity in the reticular structure of the ER marker BIP and complete loss of the reticular structure of the ER marker IPPS (Fig. 5a). Quantification of parasite populations demonstrated that over 80% of the parasites exhibited mis-localized ER (Fig. 5b). Additionally, our staining analysis revealed a dispersion of the Golgi GRASP in TBC9-mAID-depleted parasites (Fig. 5c). Notably, the dispersion of the Golgi complex could be classified into six types: single Golgi (type 1), elongated or budding Golgi (type 2), two Golgi dots (type 3), three or four Golgi dots (type 4), five or more Golgi dots (type 5), and complete diffusion of the Golgi throughout the cytoplasm (type 6) (Fig. 5c). Our quantitative analysis of the Golgi in different growth stages of parasite showed an increased tendency towards fragmentation and dispersal (Fig. 5d). Specifically, Golgi undergoing division during daughter budding exhibited the most severe fragmentation, with over 25% of parasites showing Golgi spreading into the cytoplasm (type 6) (Fig. 5d). In addition, in the absence of daughter bud formation, there was a significant increase in fragmented Golgi (type 3/4), while individual Golgi (type 1) decreased significantly (Fig. 5d). These findings suggest the potential role of TBC9

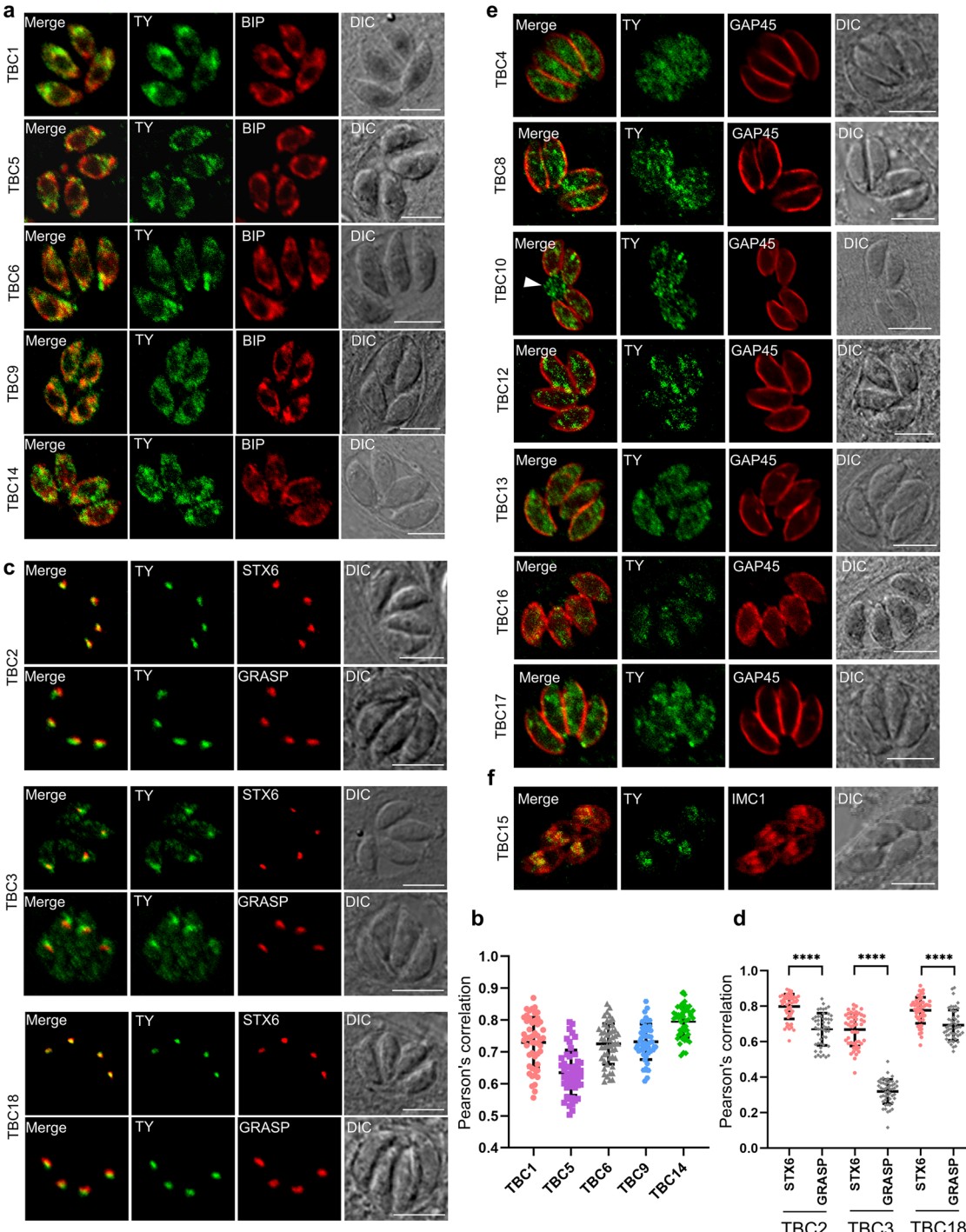

**Fig. 2 | Localization of TBC protein candidates in the parasite.** Immuno-fluorescence assays (IFAs) of endogenously epitope-tagged TBC1-10, 12–18 parasites were performed for parasites grown for 24 h. **a** Confocal co-localization of the epitope Ty fused at TBC1, 5, 6, 9, and 14 (green) with the ER marker BIP (red). **b** Statistics of Pearson correlation coefficient (PCC). The PCC values over the merged fluorescent region of TBC1, 5, 6, 9, and 14 with BIP were analyzed using the colocalization and ROI intensity analysis module in NIS software. **c** Confocal co-localization of the epitope Ty fused at TBC2, 3, and 18 (green) with the TGN marker STX6 (red) and the cis-Golgi marker GRASP (red). **d** Statistics of Pearson correlation coefficient (PCC). The PCC values over the merged fluorescent foci of TBC2, 3, and 18 with STX6 and GRASP. **e** TBC4, 8, 12, 13, 16, and 17 are localized to the vesicles of the cytosol and TBC10 are localized to the cytosol and the PV (arrowhead). The IFA analyses were achieved using anti-Ty antibodies (green), and the outline of the parasites was visualized using anti-GAP45 antibodies (red). **f** Confocal co-localization of the epitope Ty fused at TBC15 (green) with the IMC marker IMC1 (red). Data are presented as mean ± SD (n = 50 parasites), with the Mann–Whitney test used for the PCC of TBC2 and 3, and the unpaired t-test used for the PCC of TBC18 (****$p < 0.0001$). Scale bar = 5 μm.

in modulating vesicle trafficking from the ER to the Golgi apparatus, which is an early protein sorting or vesicle trafficking pathway.

Previous studies showed that TgSORTLR (Sortilin-like receptor) and TgVps35 (retromer complex protein) are required for the biogenesis of apical secretory organelles in *T. gondii* and sorting of ROP and MIC proteins[52,53]. Here, we found that depletion of TBC9 results in varying degrees of diffusion of TgSORTLR and TgVps35 throughout the cytoplasm in approximately 80% of the parasites (Fig. 5e, f). Unexpectedly, we observed

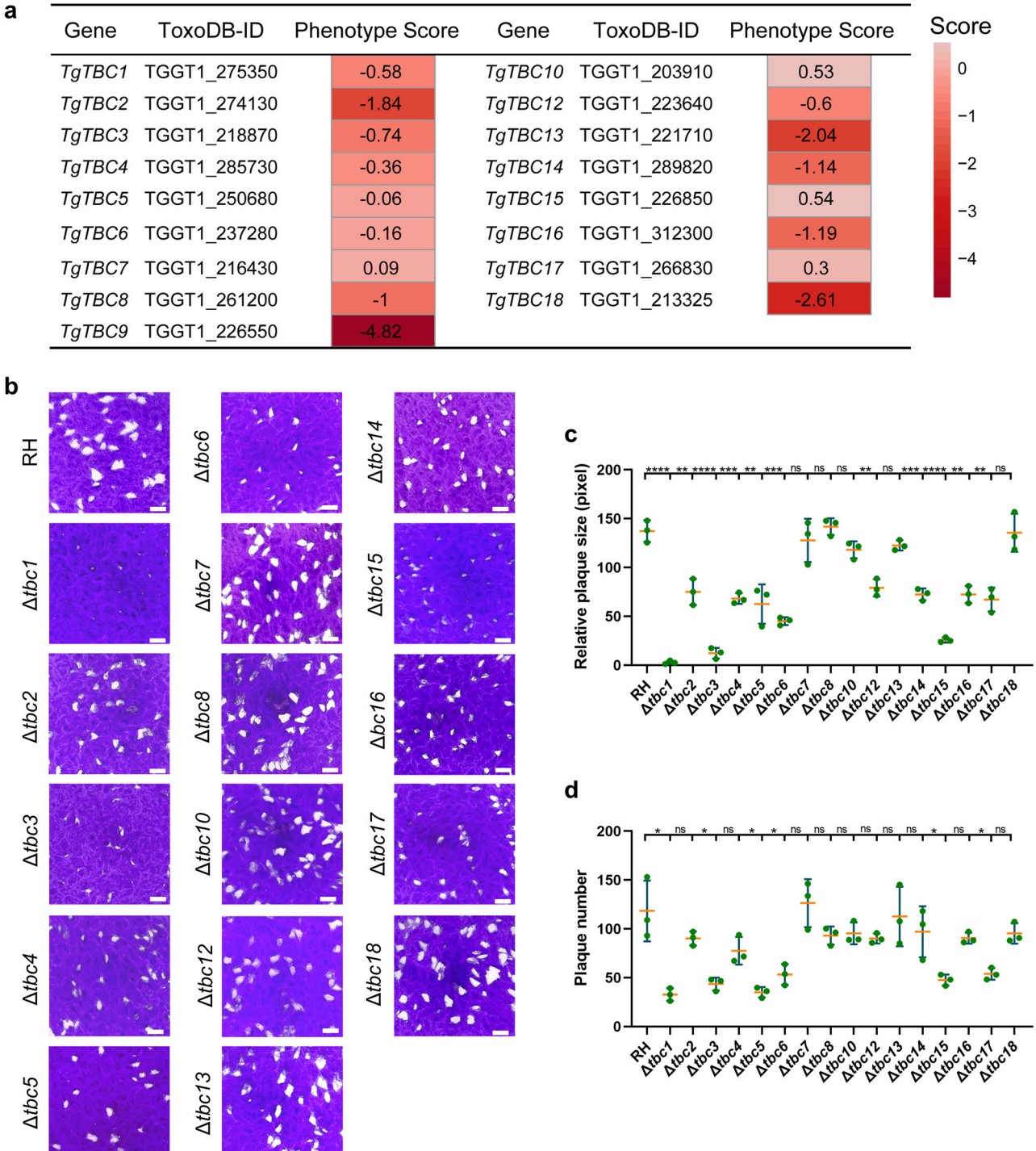

**Fig. 3 | Plaque formation of the knockout lines. a** The table lists the phenotype scores of 17 TBC genes. The phenotype scores were retrieved from a CRISPR-library screening[6]. **b** The parental line RH$\Delta ku80\Delta hxgprt$ (RH) and knockout lines of TBC1-8, 10, and 12–18 ($\Delta tbc1$-8, 10, and 12–18) were grown on HFF monolayers in 6-well plates, followed by fixation and staining with crystal violet after 7 days of culturing of parasites. The representative images for each line were shown. Scale bar = 2 mm. **c, d** The plaque size and numbers for the parasite lines examined in (**b**) were measured and plotted. Data ($N$ = 3 independent experiments; $n$ = 3 replicates) were shown with means ± SD and analyzed by the unpaired $t$ test. (ns, not significant; *$0.01 \le p < 0.05$; **$0.001 \le p < 0.01$; ***$0.0001 \le p < 0.001$; ****$p < 0.0001$).

that proteins localized to the apical organelles, such as the micronemes (MIC2[54], MIC3[55] and M2AP[56]; Supplementary Fig. 5a), the rhoptries (ROP5[57], ARO[58]; Supplementary Fig. 5b) and the dense granules (GRA7[59] and GRA12[60]; Supplementary Fig. 5c) appeared to be distributed in the regular localization. It is highly likely that depletion of TBC9 did not functionally inactivate $Tg$SORTLR and $Tg$Vps35 in the parasites, which caused undiscernible changes to the protein sorting to the apical organelles.

Subsequently, further analysis was conducted for some other organelles and proteins, such as the vacuolar compartment (VAC), the mitochondrion, and the proteins SAG1 and Actin. Our staining analysis demonstrated that the depletion of TBC9-mAID led to partial diffusion of the VAC marker CPL in approximately 50% of the parasites (Fig. 5g, h). Furthermore, over 90% of the parasites exhibited a loss of the network-like appearance of the mitochondria marker hsp60, compared to its regular localization in the non-

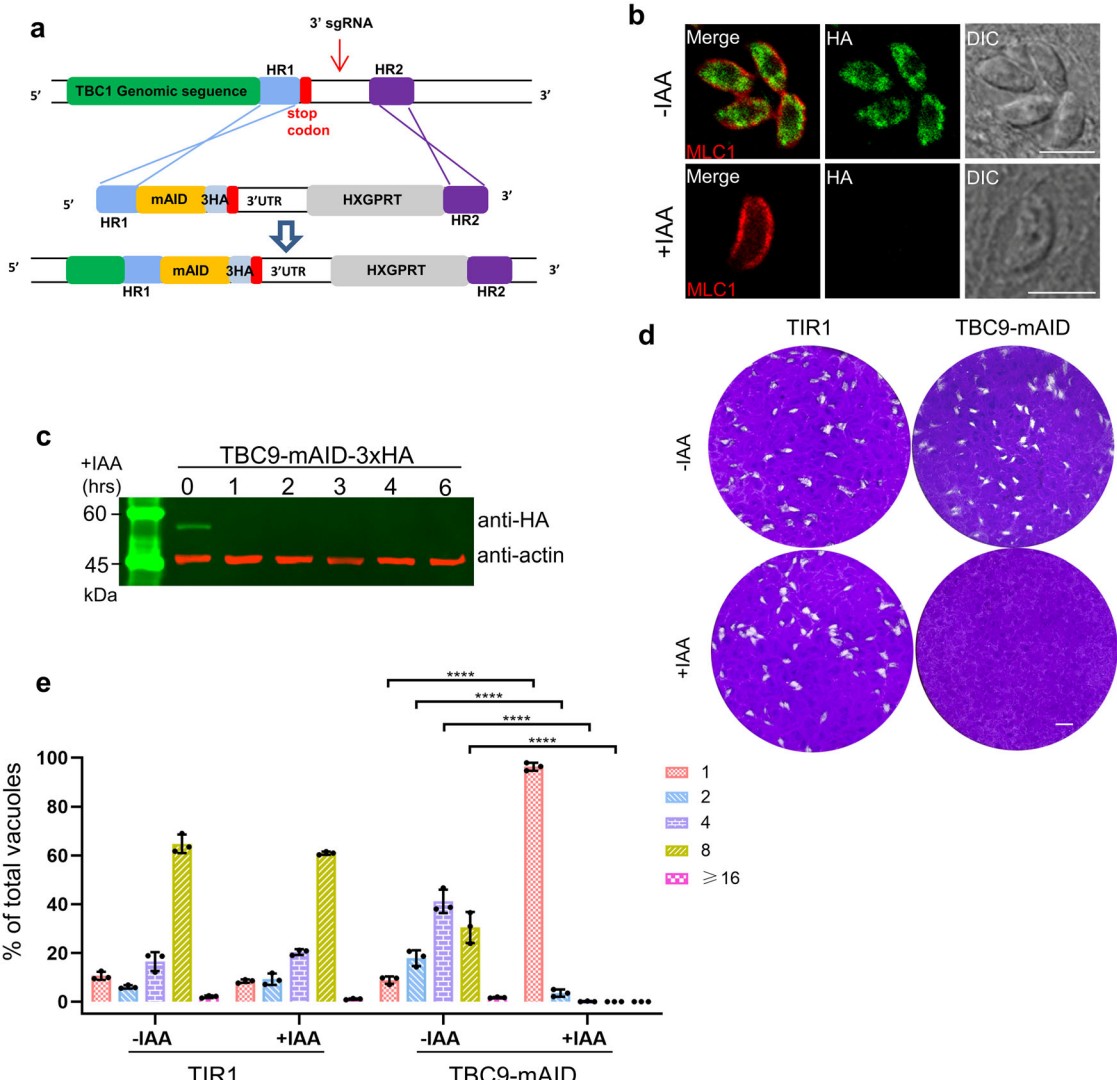

**Fig. 4 | TBC9 is essential for parasite replication. a** Schematic diagram for generation of the TBC9-mAID fusion by tagging the gene at the C-terminus with mAID-3xHA using a CRISPR/Cas9 approach in the parental line RHΔ*ku80*Δ*hxgprt*/TIR1. The homologous regions (HR) were retrieved from the upstream of the stop codon and the downstream of the 3' sgRNA targeting site, while the HXGPRT represents the resistant expression cassette. **b** The TBC9-mAID-3xHA parasite line was grown in IAA or ethanol for 24 h, followed by fixation for IFA analyses, using antibodies against MLC1 (red) and the epitope tag HA (green). Scale bar = 5 μm. **c** Western blot analysis for the TBC9-mAID-3xHA line grown for induction of auxin for different times (hrs) as indicated. Actin served as a load control. **d** Plaque formation of the TIR1 and TBC9-mAID lines. Parasites were grown on HFF monolayers in ± IAA, followed by fixation and staining with crystal violet after 7 days of culturing. Scale bar = 2 mm. **e** Parasite replication of the TIR1 and TBC9-mAID lines. Parasites were grown in ±IAA for 24 h, followed by IFA analysis of parasites by antibodies against GAP45. Vacuoles with different parasite numbers (1, 2, 4, 8, and ≥16) were counted (≥200 vacuoles for each replicate). Data (*N* = 3 independent experiments; *n* = 3 replicates) were shown with means ± SD, and were analyzed by two-way ANOVA with Tukey's multiple comparison test. (****$p < 0.0001$, and no significant differences were observed for other situations).

induced parasites (Fig. 5g, h). The membrane localization of the major glycosyl-phosphatidyl inositol-anchored surface antigens, SAG1[61], remained unaltered upon the TBC9 depletion, indicating that the integrity of the parasite pellicle was likely not to be strongly affected (Supplementary Fig. 5d). Similarly, the actin protein showed no apparent changes (Supplementary Fig. 5d), consistent with the general appearance of parasite morphology. Collectively, these findings indicate that TBC9 plays a critical role in regulating the pathways of vesicle trafficking from the ER to the Golgi apparatus in *T. gondii*.

**Depletion of the TBC9-mAID fusion impaired daughter bud formation and organelles duplication**

To gain further insights into the role of TBC9 in *T. gondii*, we examined a range of organelles upon depletion of the protein using immuno-fluorescence analyses (IFA), as demonstrated above. Firstly, we examined

the cellular structures of the inner membrane complex (the IMC, by IMC1 and GAP45), the defect of which can be easily identified by observations of the parasite morphology. Proteins for the IMC, such as IMC1, are able to stain daughter parasites and mother parasites. The IFA analyses clearly showed that after 16 h of growth, followed by 13 h of auxin treatment, the staining of IMC1 and GAP45 in parasites without daughter remains normal, suggesting that these parasites maintain their integrity in morphology (Fig. 6a). However, parasites with daughter cell budding were largely distorted with fragmented structures of the IMC (Fig. 6b). Therefore, we categorized all nuclear and IMC states into four types: single nucleus with only a mother IMC (type A); multiple nuclei with only mother IMC (type B); multiple nuclei with daughter IMC (type C); and multiple nuclei with disrupted IMC (type D) (Fig. 6a, b). Our subsequent statistical analysis revealed that approximately 13% of parasites exhibited multiple nuclei with disrupted IMC (type D) following TBC9 depletion, with no parasites

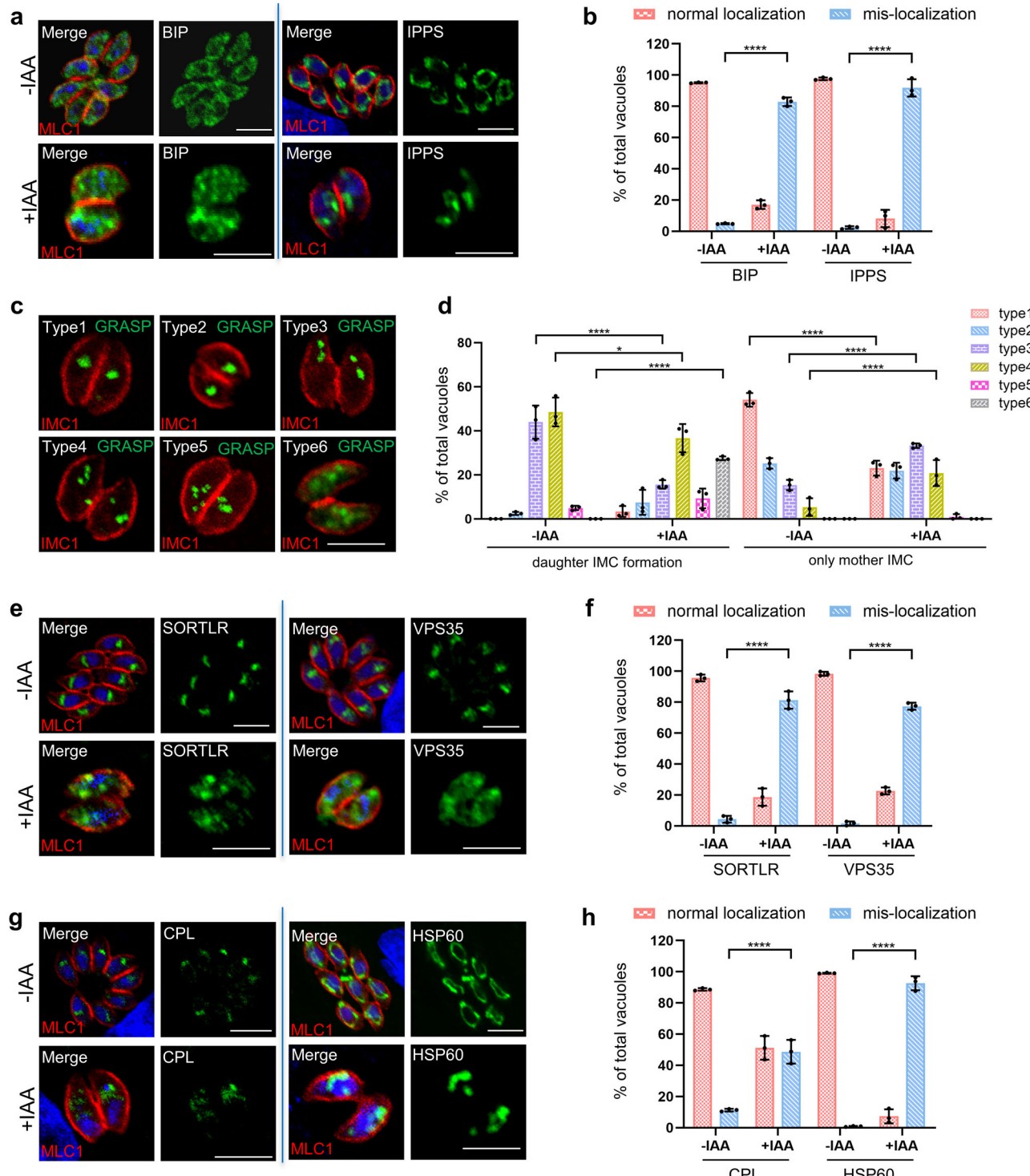

**Fig. 5 | Depletion of TBC9 affects the early vesicle transport pathway and its related proteins in _T. gondii._** Immunofluorescence analysis of normal or mis-localization of protein sorting pathway proteins and organelle marker proteins upon depletion of TBC9 in _T. gondii._ TBC9-mAID parasites were grown on HFF monolayers for 16 h, followed by ± IAA treatment for 13 h before processing for IFA analyses using the corresponding antibodies. MLC1 or IMC1 (red) for the inner membrane complex, and Hoechst (blue) dye was used to stain the nucleus. **a** BIP and IPPS staining for the ER. **b** Vacuoles with normal localization or mis-localization of ER were counted (≥200 vacuoles for each replicate). Two-way ANOVA with Tukey's multiple comparison test was performed. **c** GRASP was tagged with Ty using a CRISPR approach in the TBC9-mAID line. Six distinct types of Golgi were found for the GRASP staining. **d** Vacuoles with different GRASP types were specifically

counted during either daughter IMC formation or only mother IMC (≥200 vacuoles for each replicate). Two-way ANOVA with Sidak's multiple comparison test was performed. **e** Staining of SORTLR and VPS35 tagged with Ty in TBC9-mAID line. **f** Vacuoles with normal localization or mis-localization of SORTLR or VPS35 were counted (≥200 vacuoles for each replicate). Two-way ANOVA with Tukey's multiple comparison test was performed. **g** Staining of CPL tagged with Ty in TBC9-mAID line, and HSP60 staining for the mitochondrion. **h** Vacuoles with normal localization or mis-localization of CPL or HSP60 were counted (≥200 vacuoles for each replicate). Two-way ANOVA with Tukey's multiple comparison test was performed. Data (_N_ = 3 independent experiments; _n_ = 3 replicates) were shown with means ± SD. (*0.01 ≤ _p_ < 0.05; ****_p_ < 0.0001). Scale bar = 5 μm.

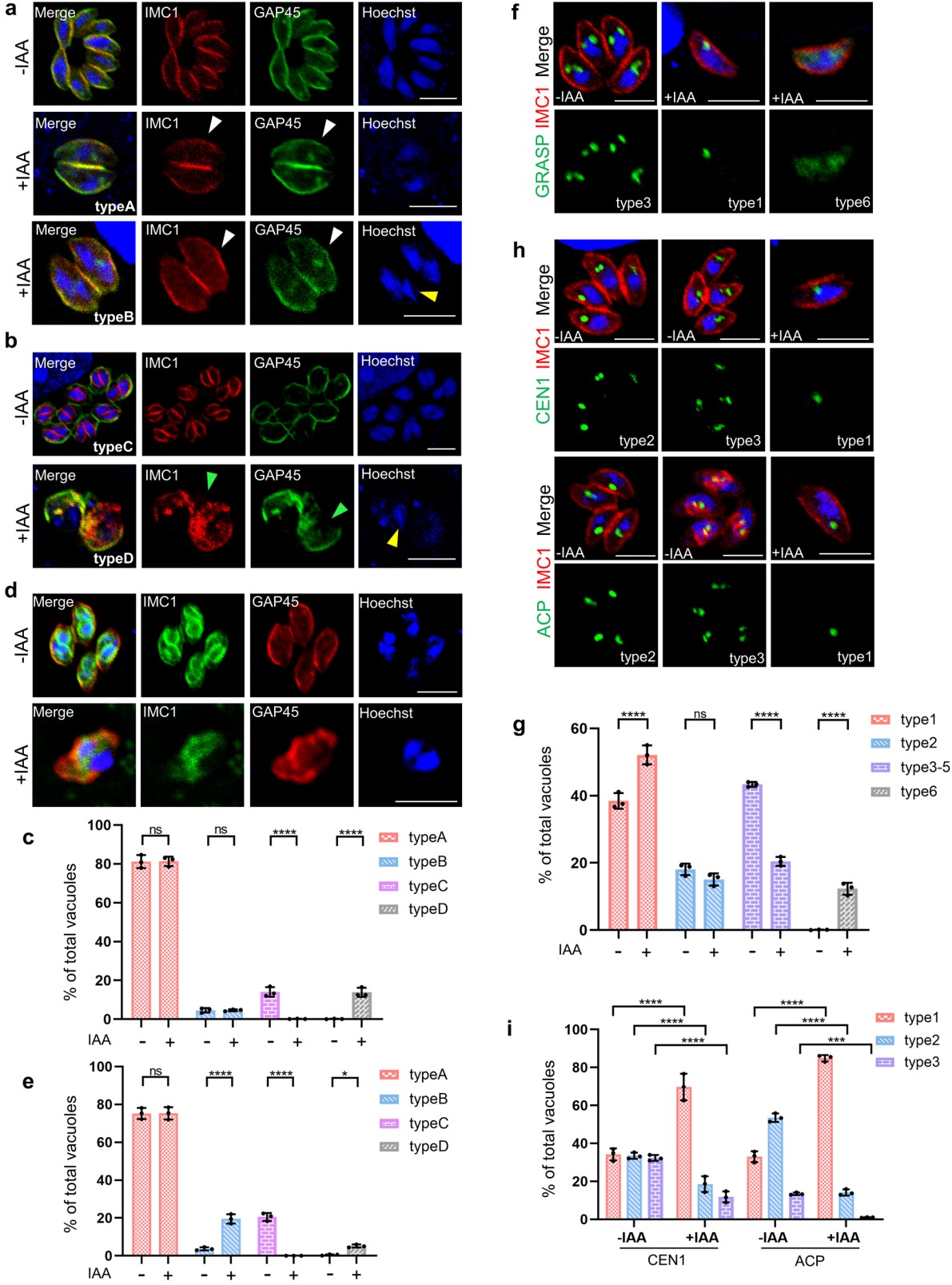

showing normal assembly of daughter IMC (type C) (Fig. 6c). Conversely, approximately 13% of non-induced parasites displayed normal budding and IMC formation (type C) (Fig. 6c). These data suggest that depletion of TBC9 disrupts the daughter IMC that is being assembled, leading to the loss of structural integrity in the mother parasite. However, it does not have a substantial impact on mother parasites that have not yet formed daughter

IMC. Moreover, the staining and quantitative results for the cytoskeleton (tubulin) also support this finding (Supplementary Fig. 6a–c). Nearly 14% of the parasites exhibited a distorted distribution of the cytoskeleton under the same induction conditions (Supplementary Fig. 6a–c). In conclusion, these data suggest that depletion of the TBC9 impacts the proper assembly of daughter cell IMC.

**Fig. 6 | Depletion of the TBC9 affects daughter bud formation and organelles duplication. a, b** TBC9-mAID parasites were grown on HFF monolayers for 16 h, followed by ± IAA treatment for 13 h before processing for IFA analyses. Hoechst (blue) dye was used to stain the nucleus. Using antibody combinations against IMC1 (red) and GAP45 (green), IFA results were obtained for morphologically normal parasites without daughter bud formation (**a**, white arrowheads) and morphologically fragmented parasites with daughter bud formation (**b**, green arrowheads), which were classified into four types (type A-D) by daughter formation and the number of nucleus (yellow arrowheads). **c** Vacuoles with 4 types in (A-D) were counted (≥200 vacuoles for each replicate). Two-way ANOVA with Sidak's multiple comparison test was performed. **d** TBC9-mAID parasites were grown for 6 h, followed by ± IAA treatment for 16 h before processing for IFA analysis. Antibodies against IMC1 (green) and GAP45 (red) were used for staining the inner membrane complex. **e** Vacuoles with 4 types (the same as in **a, b**) in (**d**) were counted (≥200 vacuoles for each replicate). Two-way ANOVA with Sidak's multiple comparison test was performed. **f** Staining of Ty-tagged GRASP in the TBC9-mAID line, and the six distinct types of Golgi are consistent with those shown in Fig. 5c. **g** Vacuoles with different GRASP types in (**f**) were counted (≥200 vacuoles for each replicate). Two-way ANOVA with Sidak's multiple comparison test was performed. **h** Staining of CEN1 for the centrosome and ACP for the apicoplast. They were classified into three types based on the replication status: not replicated (type 1), elongated with budding (type 2), and fully divided into two (type 3). **i** Vacuoles with different centrosome or apicoplast types were specifically counted (≥200 vacuoles for each replicate). Two-way ANOVA with Tukey's multiple comparison test was performed. Data ($N = 3$ independent experiments; $n = 3$ replicates) were shown with means ± SD. (ns, not significant; *$0.01 \leq p < 0.05$; ***$0.0001 \leq p < 0.001$; ****$p < 0.0001$). Scale bar = 5 μm.

In order to clarify whether depletion of TBC9 leads to an early defect in daughter cell budding (at the first division cycle), we allow the parasites to invade host cells for 6 h and then treated with auxin for 16 h. Staining of IMC1 and GAP45 revealed that depletion of TBC9 under this induction condition still resulted in IMC disruption in budding parasites (type D) (Fig. 6d). However, detailed quantitative analysis showed that the proportion of parasites exhibiting this IMC disruption was less than 5%, which is lower than parasites that underwent 1–2 rounds of division prior to induction (Fig. 6c, e). Furthermore, there was a significant increase in parasites that had completed nuclear division but had not budded daughter cells (type B) (Fig. 6c, e). These data suggests that depletion of TBC9 impedes de novo synthesis and assembly of daughter IMC, leading to difficulties in the formation of daughter buds despite completion of nuclear division. We also examined other organelles, such as the Golgi, the centrin, and the apicoplast. These organelles in the daughter parasites are generally inherited from the mother parasites during the parasite division. Using the same typing method as before (Fig. 5c), our staining of GRASP found that nearly 13% of the Golgi were also diffuse in the cytoplasm compared to the previous induction condition (16 h of growth on HFF followed by 13 h of IAA treatment) (Figs. 5d, 6f, g). On the contrary, the number of parasites containing unreplicated Golgi (type 1) increased significantly (Fig. 6f, g). Similarly, staining of centrosome maker CEN1 and apicoplast maker ACP revealed that depletion of TBC9 led to a significant increase in the number of parasites containing unreplicated type 1 centrosomes and apicoplasts (Fig. 6h, i). These findings imply that duplication of these organelles was impaired as a result of TBC9 depletion.

## Ultrastructural analysis of TBC9 depleted parasites

To verify the results obtained from fluorescence microscopy, we performed ultrastructural analysis of TBC9-depleted parasites. TBC9-mAID parasites were inoculated on HFF cells for 16 h, followed by treatment with or without auxin for 13 h. Subsequently, the intracellular parasites were fixed and subjected to ultrastructural analysis. In non-induced parasites displaying intact cellular architecture, we observed the presence of, well-organized endoplasmic reticulum, elongated lasso-like mitochondria, and closely juxtaposed Golgi stacks (Fig. 7a). Conversely, the parasites treated with auxin exhibited an abnormal morphology (Figs. 7b-f1). The ER displayed aberrant folding and disrupted network organization (Fig. 7b, c1), while the mitochondria also exhibited varying degrees of crumpling and folding (Fig. 7c2, d). Furthermore, the Golgi apparatus exhibited a loss of its characteristic tightly stacked morphology, instead presenting as loosely dispersed vesicular structures (Fig. 7e, e1), with some even extending and diffusing to larger areas within the cytoplasm (Fig. 7f, f1).

To verify the effects of TBC9 depletion on daughter budding, we allowed TBC9-mAID parasites to invade HFF cells for 6 h, followed by treatment with or without auxin for 16 h. In the absence of IAA, the daughter parasites exhibited normal budding and IMC formation (Fig. 7g). However, in the presence of IAA treatment, the daughter parasites only displayed early components of the cytoskeleton recruited during the onset of budding (Fig. 7h). There was no formation of mature apical rhoptries and micronemes, and the synthesis and assembly of daughter IMC was impaired (Fig. 7h). This impairment even resulted in the expansion and deformation of the mother parasite, with the outer membrane folding and rupturing inward towards the cytoplasm (Fig. 7i). Together, these findings validate the phenotype observations made through fluorescence microscopy.

## TBC dual-finger active sites are essential for the activity of TBC9 in the parasite

Most TBC-domain GAPs accelerate GTP hydrolysis via a shared dual-finger mechanism[31]. Since the *T. gondii* TBC9 protein contains the conserved dual-finger active sites, TBC9 is likely to operate via its active site residues in the dual-finger motif (Supplementary Fig. 1, Supplementary Table 1). To confirm this hypothesis, we attempted to complement the TBC9-mAID-3xHA line by expression of the full-length wild-type TBC9 (TBC9^wt) under the original promoter (the 5' untranslated region) of the gene. In the TBC-mAID/TBC9^wt line, TBC9-2Ty was properly localized to the ER in the parasite grown in the absence and presence of the inducer auxin (under the growth condition in auxin for 13 h) (Fig. 8a). Notably, the endogenous TBC9-mAID was able to be depleted (Fig. 8a), yet the parasites expressing TBC9^wt appeared to have normal parasite morphology, proper localization of the organelle markers IMC1, GAP45, tubulin, MLC1, HSP60 and IPPS, and similar numbers of parasites in the vacuoles (Fig. 8b, c). The observations suggested that the defects of TBC9-depleted parasites were restored by the expression of the wild-type copy of TBC9.

Subsequently, we introduced mutations to the dual-finger sites by replacement of the sites with alanine, resulting in the mutation copies: TBC9^{R74A} and TBC9^{Q101A} (Fig. 8d). Our analyses demonstrated that the mutation copies of the protein were successfully expressed in the parasite, and the localization of these copies in the parasites without auxin appeared to be ER-localized (Fig. 8e). However, upon induction by auxin, the mutation copies were diffused in the cytoplasm in the parasites depleted with the endogenous TBC9-mAID (Fig. 8e). It is noteworthy that the parasites grown in auxin appeared to be enlarged in parasite morphology, and had much less numbers of parasites in the vacuoles (Fig. 8e). We examined parasite replication by exposing them to auxin for a duration of 24 h and conducting analyses on parasite counts within individual vacuoles. As anticipated, the introduction of the wild-type TBC9 copy successfully restored the replication defect observed in the TBC9 deletion parasite (Figs. 4e and 8f). However, the expression of the point mutation TBC9 copy was unable to restore this defect (Figs. 4e and 8f). To further examine the complementation capability of the copies of TBC9 in the parasites, we then performed plaque formation assay by growing the parasite lines on HFF monolayers for 7 days. This assay demonstrated that the parasite expressing the TBC9^wt was able to form normal plaques in the presence of auxin, with no significant difference in plaque size compared to those formed in the absence of auxin (Fig. 8g, h). This result was consistent with the normal phenotypes of the proper localization of the organelle markers and parasite morphology. In sharp contrast, the parasites expressing the mutation copies of TBC9 failed to show plaques while growing in auxin (Fig. 8g, h).

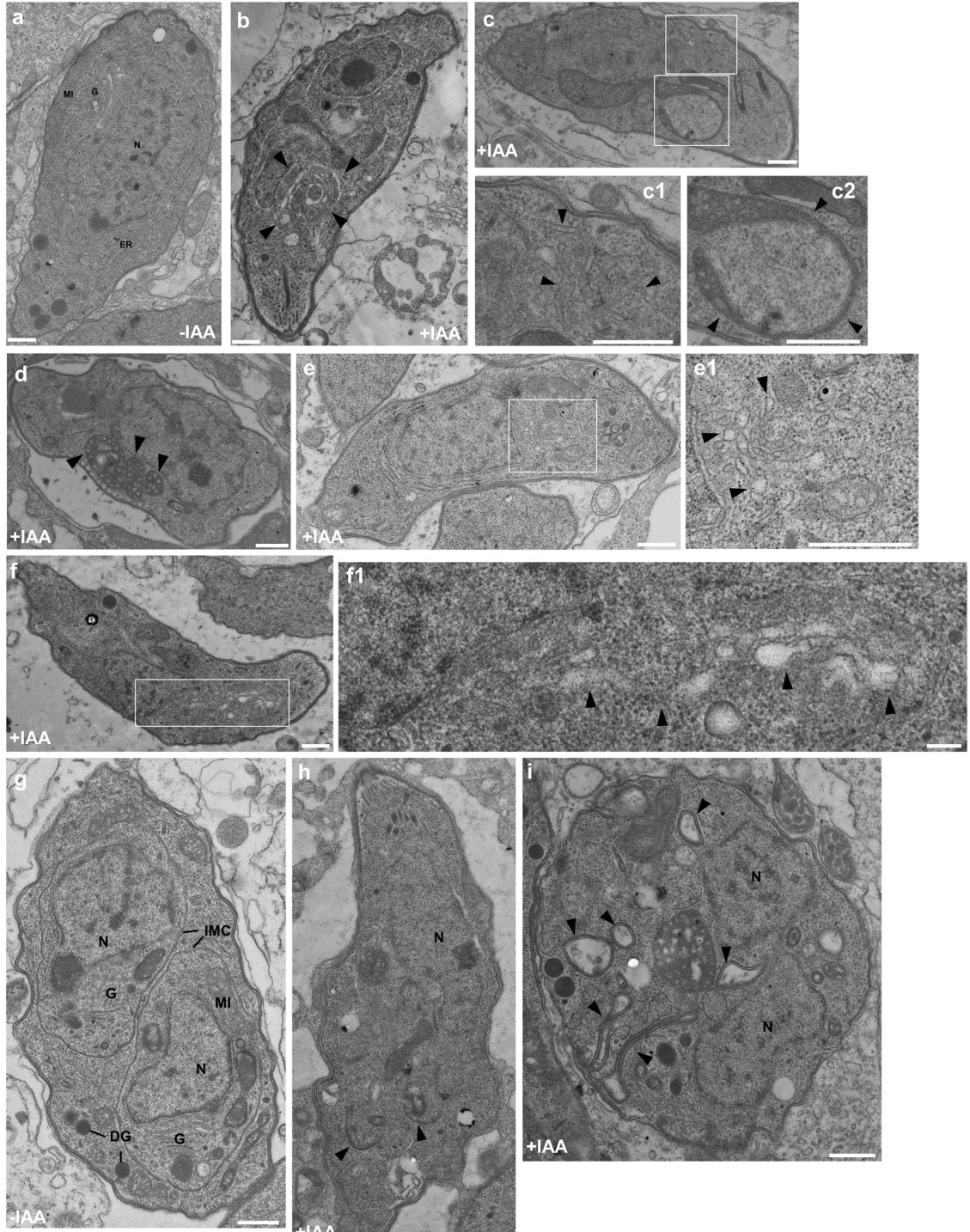

**Fig. 7 | Transmission electron micrographs of TBC9-mAID parasites treated with or without IAA.** Transmission electron micrographs of parasites grown for 16 h, followed by treatment with (**b**–**f**) or without (**a**) IAA for 13 h. **a** Longitudinal section through a control sample showing morphologically normal endoplasmic reticulum, mitochondria, and Golgi. The longitudinal sections of the sample treated with IAA show abnormally folded ER (**b** arrowheads) and disrupted ER network (**c, c1** arrowheads). Mitochondria show varying degrees of folding (**c, c2** arrowheads; **d** arrowheads). Golgi exhibits dispersed vesicular structures (**e, e1** arrowheads), some of which extend and diffuse into larger areas of the cytoplasm (**f, f1** arrowheads). Transmission electron micrographs of parasites grown for 6 h, followed by treatment with (**h, i**) or without (**g**) IAA for 16 h. The longitudinal sections of control samples reveal morphologically normal daughter buds with intact IMC, Golgi, and nuclei in the mother parasite (**g**). The longitudinal sections of samples treated with IAA show a slightly elongated parasite slice with the presence of only two conical structures representing the initiation of daughter formation, but no evidence of other daughter organizations formation (**h**, arrowheads). Spherical parasites with two nuclei and invaginated mother parasite cell membranes (**i**, arrowheads). N, nucleus; MI, mitochondria; G, Golgi; ER, endoplasmic reticulum; DG, dense granule; IMC, inner membrane complexes. Scale bar = 500 nm.

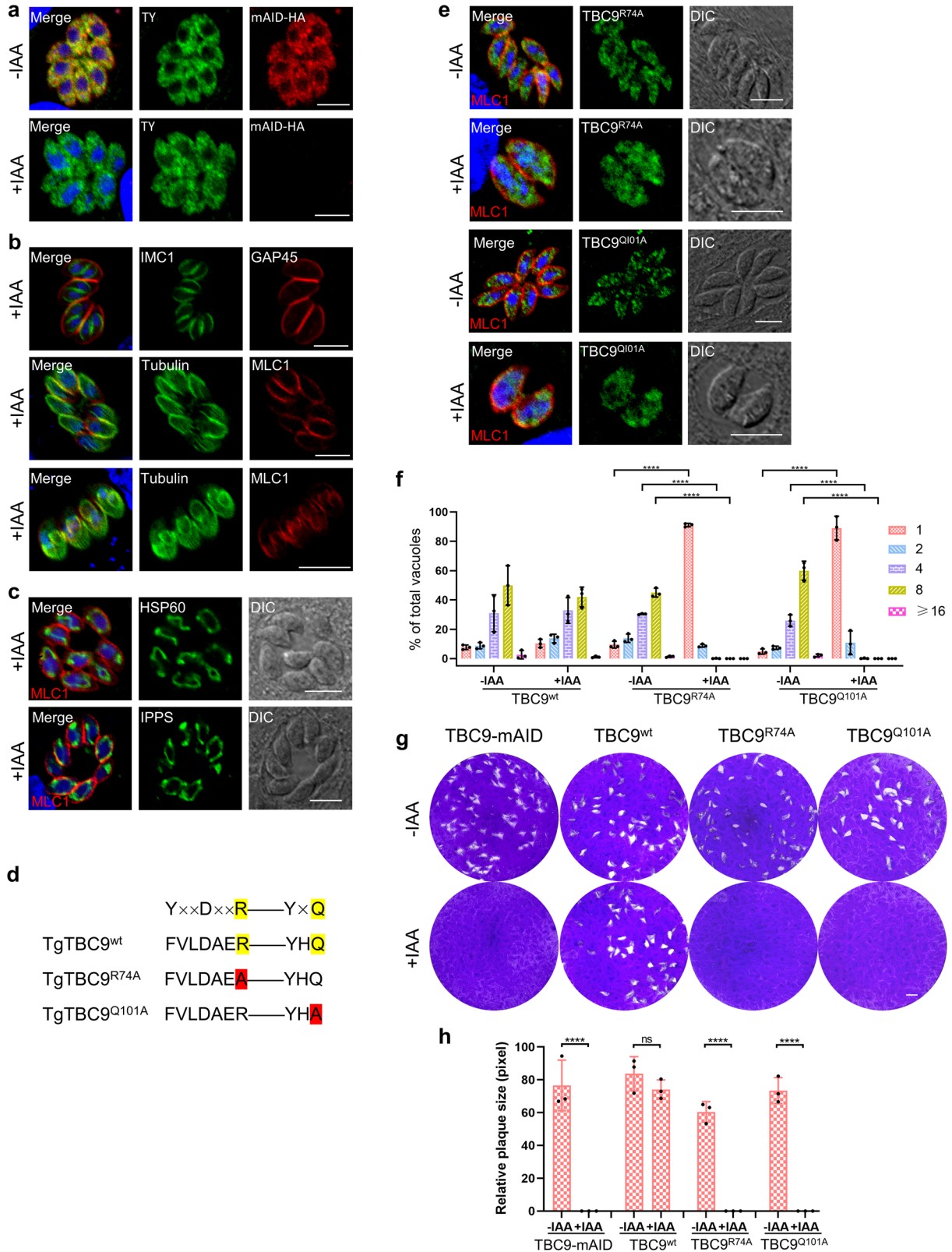

Moreover, the mislocalization of ER, Golgi, and IMC1 in parasites caused by TBC9 depletion cannot be rescued by the expression of TBC9 mutant copies (Supplementary Fig. 7). Collectively, these results demonstrated that the mutations in the dual-finger sites disrupted the protein's activity to restore the lethal defects in parasites depleted with TBC9, suggesting the key role of the dual-finger sites in the TBC9 protein.

## TBC9 interacts with Rab11A, 11B and 2 as demonstrated by pulldown assays

The TBC domain-containing proteins act as negative regulators of Rabs GTPases, catalyzing the hydrolysis of GTP by a specific GTPase in cells, and thus deactivating the Rab activity[13]. Equipped with the key role of the dual-finger sites in the *T. gondii* TBC9, we determined to identify the potential

**Fig. 8 | The dual-finger catalytic sites are essential for the activity of TBC9 in the parasite. a–c** IFA detection of TBC9-mAID-3xHA line complemented with TBC9$^{wt}$-2TY (TBC9$^{wt}$). Parasites were grown on HFF monolayers for 16 h, followed by ± IAA treatment for 13 h before processing for IFA analyses using antibodies against the proteins indicated in the images: MLC1, GAP45, and IMC1 for the inner membrane complex; Tubulin for the subpellicular microtubules[76]; HSP60 for the mitochondrion, and IPPS for the ER. Scale bar = 5 μm. **d** Diagram illustrating the TBC dual-finger catalytic sites of TBC9. The arginine and glutamine in the sites were mutated into alanine, respectively. Yellow boxes depict arginine (R finger) and glutamine (Q finger), red boxes depict residues that were mutated to alanine (A). **e** IFA detection of the TBC9-mAID-3xHA line complemented with TBC9$^{R74A}$-2TY or TBC9$^{Q101A}$-2TY.

Parasites were grown on HFF monolayers for 16 h, followed by ± IAA treatment for 13 h before processing for IFA analyses using antibodies against the TY (green) and MLC1 (red). Hoechst (blue) dye was used to stain the nucleus. Scale bar = 5 μm. **f** Parasite replication of the TBC9-mAID, TBC9$^{wt}$, TBC9$^{R74A}$ and TBC9$^{Q101A}$ lines. Vacuoles with different parasite numbers (1, 2, 4, 8, and ≥16) were counted (≥200 vacuoles for each replicate). Two-way ANOVA with Tukey's multiple comparison test was performed. **g** Plaque formation of the TBC9-mAID, TBC9$^{wt}$, TBC9$^{R74A}$ and TBC9$^{Q101A}$ lines. Scale bar = 2 mm. **h** The plaque sizes for the parasite lines examined in (**g**) were measured and plotted. Two-way ANOVA with Sidak's multiple comparison test was performed. Data ($N = 3$ independent experiments; $n = 3$ replicates) were shown with means ± SD. (ns, not significant; ****$p < 0.0001$).

interaction substrates of TBC9 in the parasite. We then generated a parasite containing TBC9 fused with a 6xHA at the C-terminus (TBC9-6HA) for a co-immuno-precipitation (Co-IP) assay. The eluted proteins from the Co-IP were examined by Western blots, together with the sample input, from which the IgG heavy chain was detected in the Co-IP eluate (Fig. 9a). Notably, a specific band of TBC9-6HA fusion in the input lane as well as the eluate lane was clearly identified (Fig. 9a), indicating that the TBC9-6HA fusion was precipitated in the Co-IP. The eluate samples were then analyzed by mass-spectrometry, which identified the fusion protein as the top hit, validating the reliability of the analysis. The enrichment analysis by comparing the fusion line to the parental line provided some Rabs candidates that exhibited the foldchange above 1 (Fig. 9b and Supplementary Data 2). Among the potential Rabs candidates detected, Rab11A was the top pulldown partner with TBC9, and the depletion of TBC9 resulted in a cytokinesis defect that closely resembled the observed phenotype in Rab11A and Rab11B defective parasites[21,22]. Moreover, previous studies have reported the localization of Rab2 and Rab18 in the ER and Golgi in the parasites. Hence, we sought to analyze the potential interaction of these proteins with TBC9. The interaction between Rab GTPases and GAP proteins (in this case, the TBC/RABGAP) is typically transient, while the interaction between TBC/RABGAP mutants, which have an inactive GAP domain, and GTP-locked Rabs is more stable, as shown in previous studies[35,62]. Therefore, to analyze the potential interactions, we expressed and purified His-tagged TBC9 and its GAP-inactive mutant TBC9$^{R74A}$, as well as MBP-tagged Rab (Rab11A, Rab11B, Rab2, and Rab18) and their GTP-locked mutants Rab$^{QL}$ (Rab11A$^{Q71L}$, Rab11B$^{Q71L}$, Rab2$^{Q66L}$, and Rab18$^{Q73L}$) in *Escherichia coli* (BL21DE3), respectively (Fig. 9c). Subsequently, we conducted in vitro pulldown assays using MBP-tagged Rab or MBP-tagged Rab$^{QL}$ as bait to specifically interact with His-tagged TBC9 or His-tagged TBC9$^{R74A}$, respectively (Fig. 9c). Our western blot analysis of the pulldown samples demonstrated that the MBP fusions, including the wild-type and mutation copies, were successfully precipitated and detected in all the tested samples (Fig. 9d). Intriguingly, the TBC9$^{R74A}$ copy, but not the wild-type of TBC9, exhibited binding to MBP agarose resin coupled with Rab11A, Rab11B, or Rab2 (Fig. 9d). Meanwhile, TBC9 of the wildtype and mutation copy was precipitated by the Rab11A$^{Q71L}$, Rab11B$^{Q71L}$ or Rab2$^{Q66L}$-MBP fusion (Fig. 9d). These results suggested that mutations in either TBC9 or Rab11A, Rab11B and Rab2 allowed for a stable interaction between TBC9 and the aforementioned Rabs. These results thus provided strong evidence supporting the interaction between TBC9 and Rab11A, Rab11B, and Rab2 in the parasite. In contrast, there was no signal of His-TBC9 or His-TBC9$^{R74A}$ detected when using Rab18 or Rab18$^{Q73L}$ as bait for pull-down, which supported the robustness of the pull-down results (Fig. 9d).

## Discussion

Apicomplexan parasites utilize a set of vesicle trafficking factors, including the Rab GTPases, to control the proper targeting and transportation of vesicles[19]. Though several Rab GTPases have been dissected on the role of vesicle trafficking in both *T. gondii* and *Plasmodium* parasites, the regulatory mechanisms governing Rab GTPases, particularly the negative regulators RabGAPs, remain unknown. In this study, we screened the potential TBC/RabGAPs in the apicomplexan model organism *T. gondii*, and identified the only essential one TBC9. Among the 17 potential TBC/RabGAPs in *T.*

*gondii*, 16 were found to localize within diverse cellular structures in the parasite, including the ER, trans-Golgi network (TGN), cytoplasmic vesicles, residual body, and daughter IMC. Nearly half of the TBC proteins were distributed in cytoplasmic vesicles, representing the most common localization pattern for most of the Rab GTPases in *T. gondii*. Consistent with previous research findings[63], three TBC proteins are localized in the plant-like trans-Golgi network (TGN), suggesting the potential involvement in the regulation of RAB GTPases present in the Golgi, including Rab1B, Rab4, Rab6, Rab11B, and Rab18[19–21]. Additionally, the TBC10 was found in the residual body (RB), a membrane-bound structure formed during tachyzoite proliferation[64]. This body encompasses the ER, mitochondria, and plasma membrane left by the dividing parent, as well as the dense granular proteins secreted by parasites and the acidocalcisome that accumulates during division, all enveloped by the intravacuolar network (IVN)[5,64,65]. The localization of TBC10 in the RB suggests that they may be retained by the parent or contribute to other functions, such as potential involvement in membrane transport between progeny tachyzoites. The subcellular localization of a few TBC proteins (TBC5, TBC8, and TBC10), differs from previous research findings[63]. This discrepancy may arise from variances in endogenous tagging and labeling sites. In apicomplexans, though the organelles appeared to be more complex due to presence of the apical organelles, the notion of simplification of the trafficking system is supported by the presence of low numbers of Rab GTPases. In here, our study provided a companion support with low number of essential TBC proteins in the parasite.

The genomes of most eukaryotes, including humans, multicellular plants, and Amoebozoa, encode fewer TBC/RABGAPs than Rab GTPases[28]. However, in *T. gondii*, the opposite is observed with 17 TBC proteins identified, in comparison with 15 Rab GTPases[19], suggesting a potential redundancy with the TBC protein family. Multiple TBC proteins localize to similar regions in the parasite, such as the ER, TGN, and cytoplasmic vesicles. This localization pattern, coupled with high phenotypic scores (indicating non-essentiality of the proteins), further supports the redundancy of protein function in the parasite[6]. For instance, TBC2 and 18 share similar localizations and have relatively low phenotypic scores of -1.84 and -2.61[6], yet, the gene knockouts grow relatively normal. We identified other TBC proteins (TBC1, 5, 6, and 14), which harbor a localization of the ER, similar to the localization of the essential one TBC9. The analysis of knockouts suggested that the ER-localized TBC proteins tend to impact plaque formation, suggesting that the ER localization of the potential trafficking factors is critical in the transport processes. In addition, cytoplasmic vesicle-localized TBC proteins are more likely to be redundant, which is likely attributed to the wide distribution of the TBC proteins in *T. gondii* (TBC3, 4, 8, 10, 12, 13, 16, 17). This localization pattern allows the proteins to compensate for the absence of the other. In our opinion, a further detailed analysis of the TBC proteins would be informative to determine the redundancy by knockout of the genes that encode proteins localized to the same location. This will contribute to a better understanding of vesicle trafficking in *T. gondii* and other related apicomplexans.

Phenotypic scores obtained from genome-wide CRISPR screens in *T. gondii* have been extensively used to assess the importance of genes in the parasite lytic cycle, facilitating researchers in identifying critical *T. gondii* genes. Consistent with the phenotypic scores, among the 17 TBC genes we

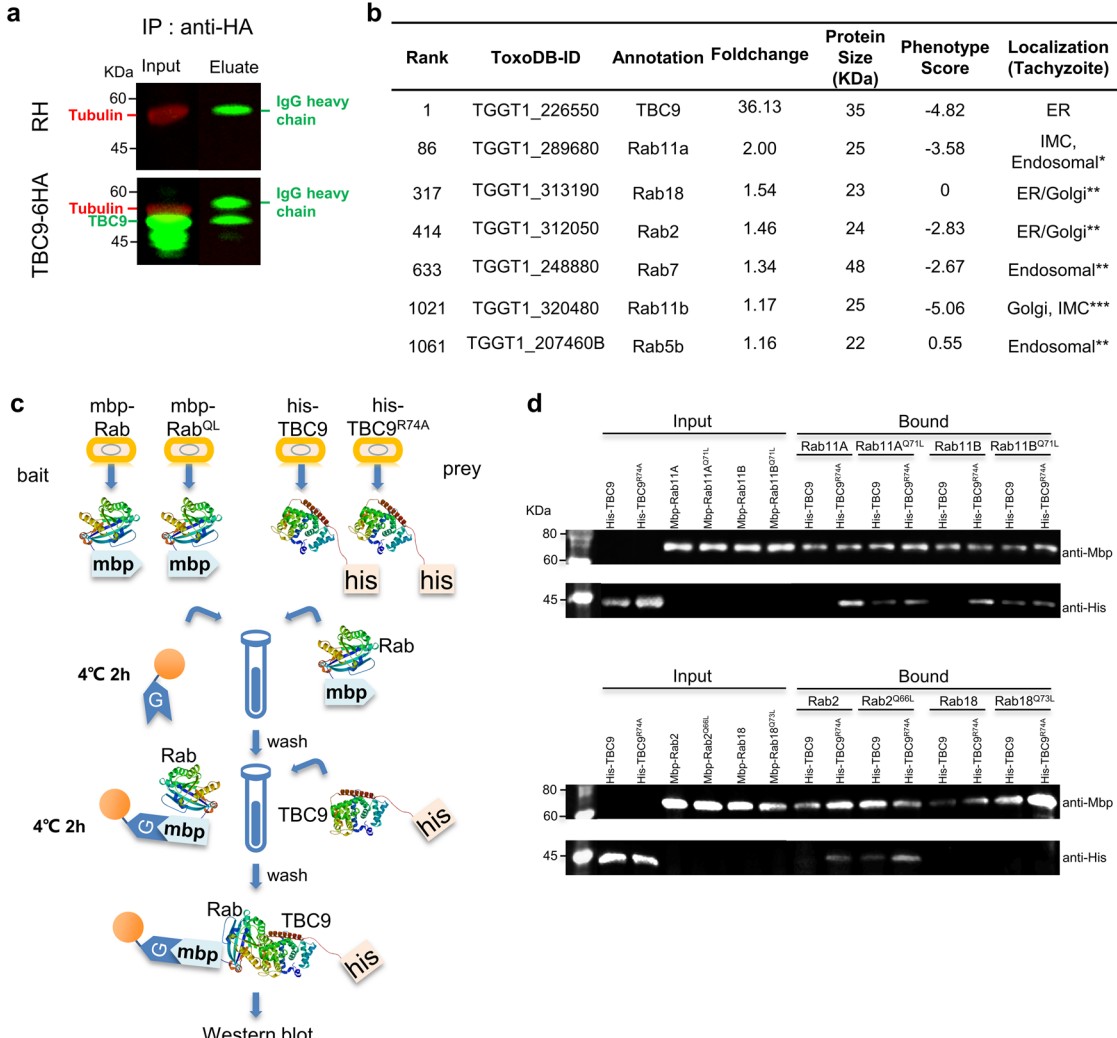

**Fig. 9 | Identification of protein interaction between TBC9 and Rab GTPases.**
**a** The parasite line TBC9-6HA was harvested for the Co-immunoprecipitation (Co-IP), and the parental line RHΔ*ku80*Δ*hxgprt* (RH) was used as control. Western blot detection of proteins purified by immunoprecipitation of TBC9 using antibodies against the epitope HA. Western blot was performed using the input samples (Input) and eluted material (Eluate). Tubulin served as a load control. **b** Mass spectrometry analysis of the proteins immunoprecipitated in (**a**). Table shows TBC9 and RabGTPase detected by the mass-spectrometry, and the hits were ranked by fold-changes obtained by comparing the mass-spectrometry area from the TBC9-6HA to that from the parental line. The phenotype scores were obtained from a CRISPR library screening[6]. The localization data for the hits can be found in previous studies (references indicated by *[22]; **[19]; ***[21]). The complete dataset can be found in Supplementary Data 2. **c** Schematic of pull-down assay. The proteins mbp-Rab,

mbp-Rab[QL] (GTP-locked mutants), TBC9, and TBC9[R74A] (GAP-inactive mutants) were expressed in *E. coli* and purified respectively. The mbp-Rab or mbp-Rab[QL] proteins were used as bait and incubated with MBP agarose resin in tubes. After washing, the resin bound with mbp-Rab or mbp-Rab[QL] was individually incubated with TBC9 or TBC9[R74A]. Subsequently, the samples were thoroughly washed and subjected to Western blot analysis. **d** Samples of MBP- Rabs (MBP- Rab11A, MBP-Rab11A[Q71L], MBP- Rab11B, MBP- Rab11B[Q71L], MBP- Rab2, MBP-Rab2[Q66L], MBP-Rab18, or MBP-Rab18[Q73L]) were incubated with MBP agarose beads, respectively, followed by mixtures with his-TBC9 or his-TBC9[R74A] in the pull-down assays. Proteins bound to the beads were analyzed by Western blot using antibodies against the His tag and MBP tag. These proteins were expressed in *E. coli* and purified using affinity standard protocols.

studied, TBC9 displayed the lowest phenotypic score and was identified as an essential gene that cannot be deleted, while the remaining 16 TBC proteins with higher phenotypic scores were successfully knocked out. However, there appears to be a notable lack of agreement between the observed plaque sizes of TBC knockout lines in our study and the fitness scores obtained from the genome-wide CRISPR screen[6]. This discrepancy may be attributed to the relatively low efficiency and potential off-target effects of genome-wide CRISPR screening, leading to false positives and false negatives. Additionally, the scoring accuracy for non-essential genes (those that can be deleted) may be more prone to error in genome-wide CRISPR screens.

In this study, we demonstrated the essential role of TBC9 in the parasite, as its knockout proved refractory to deletion in the parasite.

Depletion of this protein not only halts the growth and proliferation of the parasite but also inhibits daughter bud formation, disrupts established buds, and suppresses organelle replication. During the mitosis of *T. gondii*, the Golgi undergoes replication first, followed by centrosome duplication, and the presence of early components of the cytoskeleton initiates daughter budding[66]. Subsequently, the apicoplast moves below the centrosome and elongates as the centrosome separates[66]. As the daughter bud enters the early budding stage, Inner Membrane Complex1 (IMC1) initiates targeting of the new bud, organelles other than mitochondria start to divide and are compacted into the daughter buds[66]. The Gliding Associated Protein 45 (GAP45) and Myosin Light Chain 1 (MLC1) are recruited and assembled during the late budding stage, while the cytoskeleton of mother parasite begins to degrade, with the plasma membrane integrated onto the daughter

and itself being retained as residual body[66]. In our study, depletion of TBC9 prior to the first round of division results in inhibition of replication and division of the Golgi, centrosomes, and apicoplast. Although ultrastructural analysis demonstrates that daughter can still initiate budding, immuno-fluorescence reveals that only approximately 5% of parasites enter the budding stage, with structural disruptions observed in the IMC1, GAP45, and MLC1 proteins recruited during this stage. If TBC9 is depleted after 1–2 rounds of replication in *T. gondii*, the number of parasites entering the budding stage with disrupted IMC structures increases to approximately 13%. This suggests that the assembly of daughter IMC, which is ongoing at the time of TBC9 depletion, is also affected, preventing proper completion of final assembly and daughter membrane formation. Moreover, it was observed that the parasites did not undergo additional replication cycles (vacuoles always containing two parasites) under this induction condition. This observation provides evidence that the severe defect in the ER structure and the dispersal of the Golgi led to impaired daughter cell budding cor-related with impaired rhoptry/microneme formation in the daughter cells.

Similar to TBC/RabGAPs in mammalian cells and yeast, TBC9 also functions through the dual-finger mechanism, providing further evidence for the conservation of active sites among TBC/RabGAPs in eukaryotes[31]. Notably, there are five atypical TBC/RabGAPs in *T. gondii* that lack one or both catalytic fingers, including TBC2 (XQ), TBC5 (XX), TBC6 (XR), TBC13 (XQ), and TBC16 (XX), which may either function differently as GAPs or utilize alternative mechanisms[13,31]. Thus, we considered that this essential TBC domain containing protein TBC9 is able to regulate the fac-tors that are associated with vesicle trafficking in the parasites. Our Co-IP and pull-down data provided evidence that supports the interaction of TBC9 with RabGTPases, such as Rab11A and Rab11B. Previous studies have shown that Rab11A mediates maturation of daughter IMC and vesi-cular traffic of delivery of new plasma membrane to daughter cells, while Rab11B is involved in vesicular traffic between the Golgi and immature daughter IMC[21,22]. In the absence of TBC9 regulation, normal Rab cycling of Rab11A and Rab11B is disrupted, leading to: (1) impaired function of Rab11B in mediating daughter IMC biogenesis and inhibition of daughter bud formation; and (2) impaired function of Rab11A in controlling IMC maturation and completion of cytoplasmic division, resulting in dis-organized structures of assembling daughter buds.

On the other hand, consistent with previous studies[63], the staining assay clearly showed that the early protein sorting pathway-associated organelles, specifically the ER and the Golgi, were disrupted in the TBC9-depleted parasites, revealing that the role of TBC9 is to control the early vesicle trafficking process in the parasite. Our data, along with previous studies using immunoprecipitation (IP) and pairwise yeast two-hybrid (Y2H) experiments, provide support for the interaction between TBC9 and Rab2[63]. Rab2 was previously shown to regulate vesicular transport between the ER to the Golgi complex in mammalian cells[67,68]. In the absence of TBC9 activity, Rab2 fail to transition from its GTP-bound active state to the GDP-bound inactive state, resulting in the constant action of the GTP-bound Rab2 in the parasite. In *T. gondii*, overexpression of Rab2 was demonstrated to result in severe growth arrest, but failed to cause localization defects of the markers of the micronemes (MIC2 and MIC3) and the rhoptries (ROP2-4)[19]. Therefore, the dysregulation of the early steps of secretory protein transport by TBC9 knockdown is likely to attribute to the constant accu-mulation of active Rab2 in the parasite. In line with the information, we clearly observed in our study, that depletion of TBC9 did severely disrupt the early vesicle transport from the ER to the Golgi complex, but did not obviously affect the localization of the microneme and rhoptry markers. Therefore, we proposed that while the regulatory factor pair of TBC9-Rab2 may not play an important role in directly regulating the secretion of pro-teins in the apical organelles, excessive activation of Rab2 potentially have detrimental effects to the vesicle transport from the ER to the Golgi complex, which thus leads to parasite death. This can also explain why the division arrest is more pronounced in parasites depleted of TBC9 compared to parasites expressing dominant-negative Rab11A or Rab11B. (unable to generate multinucleated cells with more than two nuclei)[21,22].

In summary, this study elucidates the basic information of the TBC proteins encoded in *T. gondii*, providing a comprehensive understanding of these regulatory proteins in the parasite vesicle trafficking. Our findings corroborate that TBC9 is crucial for the growth of both Type I and Type II strains, and its involvement in the maintaining the biogenesis of organelles associated with the early vesicle trafficking, and the biogenesis and maturation of IMC. Collectively, TBC9 is pivotal in regulating the early protein sorting transport and the formation of IMC, which is likely to attribute to the protein interactions with Rab2, Rab11A, Rab11B in *T. gondii*, and potentially in other closely related parasites.

## Materials and methods

### Host cell culture and parasites
Human foreskin fibroblast HFF-1 cells (ATCC, SCRC-1041) were utilized as the host cells and cultured in DMEM (D5 medium) supplemented with 5% heat-inactivated fetal bovine serum (10099-141, Gibco), 100 units of penicillin-streptomycin (P1410, Solarbio Biotech) and 2 mM L-Glutamine (G0200, Solarbio Biotech) at 37 °C with 5% $CO_2$. The parental lines RH$\Delta ku80\Delta hxgprt$[69], RH$\Delta ku80\Delta hxgprt$/TIR1[48], and ME49$\Delta ku80\Delta hxgprt$/TIR1[51], referred to as RH, TIR1 and ME49/TIR1 respectively, were employed as the starting lines for generating the transgenic lines described in this study. These lines, along with the derived lines (Supplementary Data 1), were cultured in HFF-1 cells under the same media and conditions, and all remained mycoplasma-negative as confirmed by the e-Myco plus kit (Intron Biotechnology). The TIR1 and its derived AID lines were cultured in HFF monolayers with 500 mM auxin (I2886, Sigma-Aldrich) ( + IAA) or 0.1% ethanol alone ( − IAA) for phenotypic assays, as previously described[40,48].

### Antibodies
Primary antibodies, including mouse anti-HA (BioLegend, #901501), rabbit anti-HA (Thermo Fisher Scientific, #71-5500), mouse anti-his (Abmart, M30111F), and rabbit anti-MBP (Yesen, 31201ES20) were commercially purchased. Mouse anti-IMC1, rabbit anti-GAP45, rabbit anti-HSP60, rabbit anti-ACP, rabbit anti-Actin, mouse anti-MLC1, rabbit anti-STX6, and rabbit anti-IPPS antibodies were generated in our recent studies[8,23]. Rabbit anti-GRA12, rabbit anti-ARO, rabbit anti-MIC3, and rabbit anti-MIAP were obtained from other research laboratories. Other primary antibodies including mouse anti-Ty (BB2), rabbit anti-ROP5[57], mouse mAb 6D10 anti-MIC2[54], mouse mAb DG52 anti-SAG1, rabbit anti-GRA7[59] were generous gifts. Rabbit anti-BIP, rabbit anti-Tubulin, rabbit anti-Centrin1, and mouse GRASP were generated in this study. The secondary antibodies including anti-mouse antibodies conjugated with Alexa Fluors 488/568 (Thermo Fisher Scientific, A-11001 and A-11004), anti-rabbit antibodies conjugated with Alexa Fluors 488/568 (Thermo Fisher Scientific, A-11008 and A-11011), Alexa Fluor 488-Streptavidin (Thermo Fisher Scientific, S11223), fluorescent reagents conjugated with LI-COR 680 CW and 800 CW (LICOR, 926-32210, 926-32211, 926-68071 and 926-68070) were com-mercially purchased.

### Plasmid design and construction
All plasmids (Supplementary Data 1) were generated using the Basic Seamless Cloning and Assembly kit (CU201-02, TransGen Biotech) with existing plasmids as templates. The pCas9-sgRNA plasmids, used for gene tagging and knockout, contain sgRNA sequences (20 mer) targeting specific loci, which were predicted and selected using an online prediction tool as previously described[41,47]. The pCas9-sgRNA plasmids were generated using a previously described plasmid (Addgene #54467) as the template, and three fragments were amplified using three pairs of primers listed in Supple-mentary Data 1, as described in our previous study[8]. The plasmid for complementation of the TBC9-mAID line was generated by cloning the TBC9 with its 5'UTR region fragment into pLinker-2Ty-DHFR-LoxP[40] using primers listed in Supplementary Data 1. For the point mutation plasmid, the mutation sites were introduced into primers and the fragments were amplified with complementary plasmid as template (Supplementary Data 1).

## Generation of *T. gondii* lines

For endogenously tagged lines, specific CRISPR/Cas9 sgRNA 5' or 3' plasmids and a specific amplicon generated from generic plasmids by primers containing homologous regions were combined and transfected into a parental line as described in the transfection method and in our previous studies[40]. The specific CRISPR/Cas9 sgRNA 5' or 3' plasmids (Supplementary Data 1) targeting the upstream of the translation initiation code (sgRNA 5') or the downstream of the translation termination code (sgRNA 3') can efficiently produce Cas9 and sgRNA to create DNA double-strand breaks (DSB) in parasites. This transfection facilitates the integration of a tagging amplicon at the 5' or 3' terminus of a specific gene. The amplicon was generated from a generic tagging plasmid using a pair of primers containing homologous sequences of specific genes, as described in our previous study[8]. These generic tagging plasmids included pL-6Ty-HXGPRT[51], pN-6Ty-HXGPRT[70], pL-miniAID-3HA-HXGPRT[48], pL-6Ty-DHFR[8], pL-6HA-DHFR[70], and PL-AID-6TY-DHFR[8]. For generating knockout lines, we employed specific primers listed in Supplementary Data 1 to amplify fragments containing *hxgprt* from pL-6Ty-HXGPRT. Subsequently, these fragments were co-transfected with the corresponding CRISPR/Cas9 sgRNA 5' or CRISPR/Cas-sgRNA 3' plasmids into the parental line. For the complementary and point mutant lines, the plasmids were directly transfected into parasites, and the exogenous genes in the plasmids were nonspecifically integrated into the parasite genome. These transfected lines were selected by the corresponding drugs, followed by diagnostic PCR and proper localization of the corresponding proteins by IFA. If necessary, pmini-Cre was transfected to excise the resistance marker containing LoxP sites[71].

## Transfection and selection

Freshly egressed tachyzoites ($1 \times 10^7$) were mixed with the pCAS9-sgRNA plasmid (20–50 μg) and the corresponding amplicon (2–5 μg) in 200–250 μl CytoMix buffer (10 mM $KPO_4$, 120 mM KCl, 5 mM $MgCl_2$, 25 mM HEPES, 2 mM EDTA), followed by the selection, subcloning and screening methods described in a previous study[40]. The mixture was placed into a 4-mm gap BTX cuvette and electroporated using a BTX ECM 830 electroporator (Harvard Apparatus), according to a previous study[72]. Parasites were then grown in HFF monolayers and drugs were added for selection after 24 h of the electroporation. The selection was based on the appropriate concentrations of mycophenolic acid (MPA) (25 μg/ml) and 6-xanthine (6Xa) (50 μg/ml), or pyrimethamine (Pyr) (3 μM). 6-thioxanthine (6TX) (50 g/ml) was used to select the parasites transfected with pmini-Cre to remove the resistance marker. Once stable selection was achieved, IFA or PCR diagnosis was carried out to obtain positive mixed lines, and monoclonal positive lines were obtained by purifying the mixed lines if necessary.

## Indirect immunofluorescence assay (IFA)

Parasites grown in HFF monolayers on coverslips were fixed using 4% paraformaldehyde in PBS for 15 min, followed by permeabilized with PBS containing 2.5% BSA and 0.25% Triton X-100 for 20 min. After blocking with PBS containing 2.5% BSA for 30 min, the parasites were incubated with specific combinations of primary antibodies at 37 °C for 30 min. Following 3–5 washes with PBS containing 2.5% BSA and 0.05% Tween-20, the parasites were incubated with suitable secondary antibodies conjugated with Alexa Fluors-488 or −568 at 37 °C for 30 min. Prior to mounting the coverslips on glass slides, the samples were washed 3–5 times with PBS containing 2.5% BSA and 0.05% Tween-20, and the nucleus was stained with Hoechst 33258 (Solarbio C0021) for 2 min if required. Imaging of the parasites was conducted using a Nikon Ni-E microscope C2+ equipped with a DS-Ri2 microscope camera. Protein co-localization analysis was executed using the Ni-E software system installed in the NIS Elements AR.

## Plaque formation

Freshly egressed parasites were counted using hemacytometers (Hausser Scientific), and 200 parasites per well were mixed with D5 media and inoculated onto confluent HFF monolayers in a 6-well plate. For knockout lines, the parasites were cultured in D5 media at 37 °C for 7 days directly. For knockdown lines, the parasites were cultivated in D5 media supplemented with either 500 μM auxin or ethanol alone (0.1%). After 7 days (for the RH derivatives) or 10 days (for the ME49 derivatives), the 6-well plate with host cell monolayers and parasites was fixed with 70% ethanol for 15 min, stained with 0.5% crystal violet for 5 min, then rinsed with dH2O and dried at room temperature. The plaque monolayers were scanned and recorded using an HP-Scanjet G4050, and ImageJ was utilized to measure the number and size of plaques. Three independent experiments were conducted in triplicate.

## Parasite replication

The TIR1 and TBC9-mAID lines were cultured in DMEM media supplemented with 500 μM auxin or ethanol alone (0.1%) for 24 h on monolayers in 24-well plates with coverslips. The parasite and HFF monolayers were fixed with 4% paraformaldehyde for 15 min, followed by an indirect immunofluorescence assay (IFA). For IFA, GAP45 polyclonal antibodies were selected to identify the plasma membrane of parasites, and secondary antibodies conjugated with Alexa Fluor 568 were used. The coverslips were mounted using ProLong Gold Antifade Mountant without DAPI. The parasites were visualized under a Nikon Ni-E microscope C + , and vacuoles ($n \geq 200$) containing different numbers of parasites were counted. Three independent experiments with triplicates were performed, and the data represents the percentage of the total.

## Transmission electron microscopy (TEM)

The parasites grown on HFF cells were harvested and fixed with a fixation solution containing 1% paraformaldehyde and 2.5% glutaraldehyde for more than 2 h. After washing with phosphate buffer, the samples were then fixed with a 1% osmium tetroxide solution at room temperature for 2 h, followed by another wash with phosphate buffer and dehydration with acetone. Subsequently, the samples were immersed in a mixture of acetone and Spurr's resin for 12 h, followed by pure Spurr's resin for at least 8 h, and finally polymerized at 70 °C for 14 h. Ultrathin sections of 60–80 nm thickness were obtained using a Leica EM UC7 ultramicrotome and placed on single-hole grids coated with Formvar membrane. The ultrathin sections were stained with 2% uranyl acetate and lead citrate, and observed and photographed using a Hitachi HT7800 transmission electron microscope.

## Co-Immunoprecipitation (Co-IP) and western blots

Freshly egressed tachyzoites ($1 \times 10^9$) from TBC9-6HA and RH$\Delta ku80\Delta hxgprt$ lines were collected and lysed on ice for 40 min using Native lysis Buffer (Solarbio R0030) supplemented with PMSF protease inhibitor. The lysates were centrifuged at 21, 130 g for 15 min at 4 °C, and input samples were collected. The supernatants were then incubated with Anti-HA Magnetic Beads (BeyoMag™ P2121) overnight at 4 °C on a rotating shaker. After collecting the unbound samples, the magnetic beads were washed three times with 500 μl of 1XTBS. The magnetic beads were resuspended in PBS with 5xLaemmli sample buffer for the detection of bound samples. The purified samples were subjected to SDS-PAGE and silver staining prior to mass-spectrometry analyses. For Western blots, collected parasites were resuspended in PBS and mixed with 5xLaemmli sample buffer. Protein samples were heated at 95 °C for 10 min and were separated on 10% acrylamide gels, followed by blotting using a BioRad wet-blotting system. Proteins were detected using specific primary antibodies and secondary antibodies conjugated with LI-COR 800CW or 680CW reagents. The membranes were visualized using a Bio-Rad ChemiDOC MP imaging system.

## Proteomic analysis of Co-IP samples

Protein digestion was performed using the FASP method with modifications[73]. Briefly, 100 g of protein was subjected to reduction, alkylation, and washing steps to remove stains and SDS. The samples were digested with trypsin at 37 °C overnight, and the digested peptides were collected and diluted with 0.1% formic acid (FA) for nanoLC-MS analysis. The nanoLC separation was achieved with a Waters (Milford, MA, USA)

nanoAcquity nanoHPLC with a gradient elution program. Nanospray ESI-MS was performed on a Thermo Q-Exactive high-resolution mass spectrometer (Thermo Scientific, Waltham, MA, USA). Raw data from the mass spectrometer were preprocessed by PeaksOnline engine for peak picking[74,75]. The control line RHΔ*ku80*Δ*hxgprt* and TBC9-6HA lines were analyzed in parallel with two biological replicates. Resulting spectra were searched against combined database of *T. gondii* (http://ToxoDB.org, release 53) using the PeaksOnline engine. The search parameters were as follows: Fixed modifications: Carbamidomethyl (C), Variable modifications: Oxidation (M) Enzyme: Trypsin, Maximum missed cleavages: 2, MS mass tolerance: 10 ppm, MSMS mass tolerance: 0.02 Da. The foldchanges were calculated by comparing the mass-spectrometry areas from the TBC9-6HA to that from the parental line (TBC9 average Area/RH average Area).

### Pull-down assay

The *T. gondii* cDNA fragments encoding the full-length TBC9 protein were cloned into pET28a vector (Thermofisher Scientific) using specific primers to generate the corresponding expression plasmids (Supplementary Data 1). Using this plasmid as a template, the point mutation expression plasmid pET28a-TBC9$^{R74A}$ was constructed using appropriate primers. The *T. gondii* cDNA fragments encoding the full-length Rab proteins were cloned into the pMBP vector to generate the pMBP-Rab expression plasmids, respectively. The point mutation expression plasmid pMBP-Rab$^{QL}$ were generated using suitable primers and the corresponding pMBP-Rab plasmid as templates. These proteins were expressed in the *E. coli* BL21DE3 (Transgene Biotech, CD601-02) and purified by His-Tagged Protein Purification Kit (Cwbio, CW0894S) or MBPSep Dextrin Agarose Resin (Yesen, 20515ES08). The protein concentration was determined by Pierce BCA Protein Assay Kit (Thermofisher Scientific, 23227). For the pull-down assay, 4 μg of purified MBP-RAB or MBP-RAB$^{QL}$ protein and 500 μL of MBP protein binding buffer (containing 1 mM DTT and 1 mM PMSF) were incubated with 20 μL of MBP agarose resin prewashed with MBP protein binding buffer on a rotary shaker at 4 °C for 2 h. After centrifugation at 500 g for 3 min at 4 °C, the supernatant was discarded and the resin was washed three times with 1 mL of MBP protein binding buffer. Then, 4 μg of purified his-TBC9 or his-TBC9$^{R74A}$ protein and 500 μL of his protein binding buffer were added to the resin and incubated at 4 °C for 2 h with gentle agitation. After centrifugation at 500 g for 3 min at 4 °C, the supernatant was discarded and the resin was washed enough times with 1 mL of his protein binding buffer containing 0.1% Triton X-100. The resin was resuspended in PBS and mixed with 5xLaemmli sample buffer, boiled for 10 min to prepare the sample, and analyzed by Western blotting for the pull-down result.

### Antibody preparation

To create the corresponding expression plasmids (Supplementary Data 1), cDNA fragments encoding full or partial polypeptides of BIP, Tubulin, Cen1, and GRASP from *T. gondii* were cloned into pET28a using primers. The proteins were expressed in *Escherichia coli* BL21DE3 (Transgene Biotech, CD601-02) and purified by HisPur Ni-NTA rotary columns (Thermofisher Scientific, 88225). Following the manufacturer' protocols, the recombinant proteins prepared by Inject Freund's complete adjuvant (Sigma-Aldrich, F5881) was used to produce antiserum in mice or rabbits. BALB/C mice and New Zealand white rabbits were used in this study, following the guidelines and regulations provided by the Veterinary Office of China Agricultural University (Issue No. AW11402202-2-1). The antisera were subjected to a series of dilutions for evaluation via IFA and western blots. Further assessment was conducted based on the intraparasitic localization and the molecular weights of the corresponding proteins in *T. gondii*.

### Bioinformatic analysis

To identify proteins containing the TBC domain in *Toxoplasma gondii* GT1, we conducted a search on TOXODB using the InterPro (https://www.ebi.ac.uk/interpro/) ID IPR035969 (Rab-GAP-TBC domain superfamily) as the query. For multiple sequence alignment analyses, protein sequences were retrieved from ToxoDB and aligned using the L-INS-I algorithm in Mafft

software (v7.490, Kyoto, Japan). The alignment results were displayed using Genedoc software (version 2.7). For phylogenetic analyses of TBC9, protein homologues were initially downloaded from OrthoMCL DB (https://orthomcl.org/), and the most representative species were selected. Subsequently, the amino acid sequences of the selected species were aligned using the L-INS-I algorithm in Mafft software, and spurious sequences or poorly arranged regions were automatically removed using trimAL software (https://vicfero.github.io/trimal/). Finally, The Le-Gascuel model was used to construct the phylogenetic tree through FastTree (v2.1, Berkeley, CA, USA), and the resulting maximum likelihood tree was displayed using the Chiplot online portal (https://www.chiplot.online/).

### Statistics and reproducibility

Data in this study were collected in a blinded manner and analyzed using Graphpad Prism software (version 8.4.0). Two-way ANOVA with Tukey's or Sidak's multiple comparison was performed on data with normal distribution and two independent variables. The *t* test was used to compare two groups of independent data. Specifically, the unpaired T-test were performed on data with normal distribution, while the Mann–Whitney test was performed on data without normal distribution. $P < 0.05$ was considered statistically significant. Statistics for specific experiments, including the number of replicates (n), trials (N), and statistical tests performed, are provided in the figure legends or associated method details.

### Reporting summary

Further information on research design is available in the Nature Portfolio Reporting Summary linked to this article.

### Data availability

The proteomic data reported in this paper have been deposited in the OMIX, China National Center for Bioinformation/Beijing Institute of Genomics, Chinese Academy of Sciences (https://ngdc.cncb.ac.cn/omix: accession no. OMIX005138). Minimally processed data of the proteomics are available in Supplementary Data 2. The source data behind the graphs in the paper is available in Supplementary Data 3. Requests for reagents can be directed to the correspondent author.

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

## Acknowledgements

We appreciate the assistance provided by the biological mass spectrometry laboratory in College of Biological Sciences, China Agricultural University. We thank Prof. Vern Carruthers for the kind gift of RHΔ*ku80*Δ*hxgprt* line, Prof. David Sibley for antibodies against MIC2, ROP5, GRA7, and for the TIR1 lines, and Prof. Philippe Bastin for the hybridoma line BB2 for generation of monoclonal antibodies against the epitope tag Ty. This research was supported by the National Key Research and Development Program of China (Grant Nos. 2022YFD1800202A2) to S.L.

## Author contributions

Conceptualization, S.L. and M.S.; Supervision and funding acquisition, S.L.; Investigation, M.S. and T.T.; Methodology, M.S. and S.L.; Software, M.S. and K.H.; Writing—original draft, M.S.; Writing—review and editing, S.L. and M.S. All authors have read and agreed to the published version of the manuscript.

## Competing interests

The authors declare no competing interests.
