## [Peer Review File · Communications Biology]

Reviewers' comments:

Reviewer #1 (Remarks to the Author):

In their paper titled " Profiling of TBC domain-containing proteins reveals that the only essential protein TBC1 regulates the Early Vesicular Transport Pathway in *Toxoplasma gondii*," submitted on October 25, 2023, Sun et al investigate the importance of TBC domain proteins in the apicomplexan parasite *Toxoplasma gondii*. TBC domain proteins are known to be Rab GTPase regulators in eukaryotes. Rabs have been demonstrated to play a crucial role in the *Toxoplasma* lytic cycle. Studying how these Rabs are regulated is particularly interesting, especially given that very little was known about them.

Using Cas9 to tag and then disrupt the genes (straight KO, then Auxin for TBC1, which could not be obtained directly), the authors examined 17 TBC proteins identified using ToxoDB and bioinformatics analysis. Focusing on the most essential protein, TBC1, they conducted a comprehensive investigation of the impact of the gene's depletion using the auxin degradation system. They notably deeply investigate the state of the parasite's organelles. They also carried out point mutations on the dual-finger part of the TBC domain.

The results showed that TBC1 is crucial for the trafficking of vesicles from the endoplasmic reticulum (ER) to the Golgi apparatus. Finally, using pulldown assays, they demonstrated that TBC1's primary partner is Rab2, and that TBC1 is also essential in Type 2 strains.

Major issue:

The paper is well-structured, with well-presented results that require only minor modifications for clarity. However, there is a major ethical issue that needs attention.

As mentioned, the paper was submitted on October 25, 2023. Since May 29, 2023, a paper titled "*Toxoplasma gondii* encodes an array of TBC-domain containing proteins, including an essential regulator that targets Rab2 in the secretory pathway," authored by Quan et al, has been available on BioRxiv (<https://www.biorxiv.org/content/10.1101/2023.05.28.542599v1.full>).

The striking homology between these two papers, from the technics used to the organization of the content, is hard to overlook. On ToxoDB, all the TBC proteins mentioned in the BioRxiv paper were already assigned gene IDs in user comments since May 29. TBC1 in the submitted paper is identified as TBC9 in the BioRxiv paper.

The lack of any reference to the BioRxiv paper in the submitted manuscript is concerning. At a minimum, a citation to the BioRxiv paper should have been included.

Have the authors of the BioRxiv paper have been contacted regarding this overlap? If the similarity between the two papers is purely coincidental, a back-to-back publication strategy would have been preferable to uphold good scientific practice, especially given the relatively small field of *Toxoplasma* research. The BioRxiv paper should already be under revision somewhere.

This matter need clarification, because in it state I can not approve any publication.

Minor revisions:

After addressing this disturbing issue, I will proceed with my comments on the article

Experiments:

In the results section, the last paragraph discusses the evaluation of the impact of TBC1 knockout in type 2, even though the authors have already clearly demonstrated its essential role in type 1. It would be more interesting to assess bradyzoite differentiation in type 2 rather than solely examining the KO's impact. If no additional analysis is planned for type 2, it might be sufficient to mention this aspect briefly when discussing the generation of their mAID strain in the section titled 'TBC1 is essential for parasite replication,' rather than dedicating a whole separate section to it

Text revisions:

As mentioned earlier, TBC1 corresponds to TBC9 in the BioRxiv paper. Since ToxoDB has already been annotated with the 18 TBC proteins reported in the BioRxiv since May 2023, it would be advisable to standardize the nomenclature to avoid confusion in the future.

On line 199, the conclusion that 'most of the TBC proteins are localized in cellular structures associated with protein secretion or protein sorting pathways' appears to be somewhat overstated. While 8 out of 17 are indeed located in the ER/Golgi area, 9 are distributed in the cytoplasm, residual body, or remain unlocalized.

On lines 244-256, the explanation of the observed phenotype for the Inner Membrane Complex (IMC) may benefit from clarification. The author could clarify that, in 87% of cases, no impact is observed on the IMC, but a small proportion of parasites exhibit IMC distribution issues, accounting for 13%.

On line 305, I am wondering about whether the parasites with diffuse Golgi correspond to those with diffuse IMC1. Since both are observed in the same proportion (13%), it could be interesting to perform a dual-label IMC1-Golgi.

Lastly, on line 592, there seems to be a reference error (Error! Bookmark not defined).

Figure Comments:

In general, there are some key quantifications that are missing and should be included in the figures. These could include data on the proportion of parasites with disrupted organelles due to TBC1 KO, as well as more detailed information on fine co-localization.

Regarding Figure 2, the co-localization figures in Figure S2 appear to be more informative than those presented in Figure 2. It may be beneficial to consider swapping them, for the protein with a well-defined localization.

For Figure 3, there seems to be a small issue with the representative pictures for TBC9 and TBC12, as they don't align with their respective area analysis. In the picture, TBC9 plaques appears larger than TBC12, whereas the area analysis suggests otherwise.

In Figure 4, the TBC1 m-AID plaque without IAA appears smaller than the Tir1 control. It would be helpful to provide some commentary or explanation for this observation.

Figure 5: It might be a good idea to group all the IFA data from both Figure 5 and Figure S5 together. This

could make it easier for readers to follow and compare the results.

For Figure 6, adding some quantification analysis could enhance the understanding of the impact of the different point mutations.

In Figure 7, the Co-IP figure appears somewhat complex. Creating a small schematic to indicate that Rab 2 and 18 are tagged with Mbp, TBC1 with His, and that different pull-down conditions are applied to both the wild-type (WT) and point mutation samples could significantly improve reader comprehension.

Reviewer #2 (Remarks to the Author):

Ming Sun et al explore the function of the TBC family of Rab GTPase Activating Proteins in *Toxoplasma* tachyzoites. By first screening the 17 member TBC family for expression, localization, and essentiality, they were able to focus on a single essential TBC protein called TBC1. They determined that TBC1 is an ER protein that is critical for regulating protein sorting from the ER to the Golgi. Loss of TBC1 resulted in morphological defects, organellar defects, and lethality due to aberrant replication. Furthermore, the authors determined that TBC1 requires a dual finger catalytic site and that may interact Rab proteins such as Rab2, as expected for a TBC protein but not greatly supported by their data. Altogether, the authors resolve some of the functional redundancy within an important gene family (TBC) and provide some mechanistic insights into the essential role of TBC1 in tachyzoites. This study is novel and opens the door for further investigation of Rab-dependent processes in *Toxoplasma* that could be targeted by drugs to treat *Toxoplasma* infections.

Major concerns

1. There seems to be very little agreement between the TBC knockout phenotypes generated here (Fig. 3) and the CRISPR/Cas9 KO screen fitness scores reported by Sidik et al. 2016 ref 6. Specifically referring to TBC2,3,4,5,9,10,12,13,16. Meaning those with low CRISPR fitness scores formed larger plaques than those with higher CRISPR fitness scores, opposite of expectations. There is so much inconsistency that it casts some doubt on the identities of the mutants shown. Please double check that the data and representative images perfectly match the mutants. Since genome-wide CRISPR screens have false positives and false negatives due to inefficiencies and off-targets, there may be a reasonable explanation for the inconsistencies. Please dedicate a paragraph in the discussion to explain this discrepancy.

2. The TBC1-6HA co-IP data shown in Figure 7B is difficult to interpret because the parental negative control data is not included in Supplement Dataset 1. There should be 2 replicates for control, 2 replicates for the test, averaged data with fold change and P values. The link between TBC1 and Rab2 is extremely spurious based on the co-IP data (I believe it ranks 414 based on fold enrichment). A better presentation of the data file and more accurate description of the results is needed. Any additional method to show TBC1 interaction with Rabs by TurboID or reciprocal co-IP with western blotting would be more convincing than the recombinant protein data shown in Fig. 7C.

Minor concerns

1. Title: The title is misleading. Other TBC proteins may also regulate early vesicular transport. Consider revising to "Profiling of TBC domain-containing proteins reveals TBC1 as an essential regulator of the

- early vesicular transport pathway in *Toxoplasma gondii*" or further clarify the original title.
2. Line 21: write out TBC in abstract.
 3. Line 23: Functional profiling of these proteins in tachyzoites...
 4. Line 41: Delete inciting
 5. Line 43: Delete become established and
 6. Line 48: Change operations to manipulation
 7. Line 133: It would be simpler to just say you identified 17 TBC proteins by searching ToxoDB using the Interpro ID IPRO35969 Rab-GAP-TBC domain superfamily.
 8. Line 212: Period after BIP.
 9. Lines 245-251: Please better connect the treatment conditions and controls with the observed phenotypes.
 10. Figure 1: How were the TBC genes assigned numbers 1-17?
 11. Figure 2: Do the TBC protein localizations match the Hyper-LOPIT study? Barylyuk et al. 2020 CHM.
 12. Figure 3B: It is unclear what Plaque Area % means. For instance, what does 7% plaque area mean for RH control? Should % simply mean mm² or pixels². Please explain why unpaired T-test and Welch's T-test were used and which samples they were used for.
 13. Figure 4B/Lines 229/230: It appears the mAID-3HA tag on TBC1 was responsible for an innate hypomorphism (smaller plaques). This should be acknowledged in the results at lines 229/230.
 14. Figure 4F-I: Label panels that are considered normal (F-G) or fragmented (H-I) in the figure.

Reviewer #3 (Remarks to the Author):

In this study, Sun et al. examined the role of the TBC domain containing protein family in *Toxoplasma gondii*. By bioinformatic search, they identified 17 genes encoding TBC proteins, among which only TBC1 (out of 17 identified TBC proteins) was found to be essential for the life cycle of the parasite in fibroblasts. Based on localization study after endogenous tagging of the protein, the authors concluded that TBC1 is localized in the ER. Inducible knock-down approach showed that TBC1 is essential for parasite replication. Examination of different parasite organelles by immunofluorescence showed that the pellicule (tubulin, actin cytoskeleton and IMC), the Golgi, the mitochondria, the ER and the VAC (but not the rhoptries and micronemes) morphology is altered upon TBC1 depletion. The authors concluded that TBC1 regulates trafficking between the ER and the Golgi.

The study is interesting for the *Toxoplasma* community providing new knowledge on the regulation of vesicular trafficking in this parasite via the identification and description of novel GAPs. However, while the role of TBC1 in parasite replication through its conserved dual-finger active sites (point mutation approach) has been clearly demonstrated, major conclusions can not be drawn from the results provided in this study :

- 1) The localization of TBC1 in the ER can not be concluded from the provided images, notably due to a lack of resolution and saturated signals. A major part of the TBC1 signal does not co-localize with the ER.

2) The role of TBC1 in regulating vesicular transport between the ER and the Golgi can not be concluded due to pleiotropic effects on most of the parasite organelles, lack of clear TBC1 localization in the ER, only very partial effect on the Golgi apparatus (not explained) and what seems to be a very drastic defect in early phase of daughter pellicle formation. Loss of parasite polarity in absence of daughter bud formation leads to pleiotropic effects on most of the parasite organelles.

3) The authors partly based their conclusions on the interaction between Rab2 and TBC1, although Rab11A was the top pulled-down partner of TBC1 and indeed, the TBC1 depleted parasites share many defects observed in Rab11A/B defectives parasites, previously described in the literature.

Specific comments :

Figure 2 : How do the authors explain the localization of some TBC in the residual body ?

Line 165 : Immunofluorescence assays (IFA) showed that the TBC1, 3, 10, 13 and 16 were primarily located in the cytoplasmic region or the endoplasmic reticulum (the ER) (Figure 2A). Colocalization studies with a ER marker cis-IPPS (decaprenyl diphosphate synthase) (43) showed at least a partial co-localization with the ER was observed for these TBC proteins (Figure S2D).

Do the authors consider that all these cited TBC localize in the cytosol « or » the ER, or rather that all these TBC localized in the cytosol AND the ER ?

In Fig S2, please provide better quality images including improved resolution, zoom in the ER region as well as a non-saturated signal for the ER marker cis-IPPS.

Figure 3 : TBC10 seems to be also essential. How do the authors successfully maintain those parasites in culture ? the plaque area suggests that they do not replicate.

Figure 4 : Please place the results of figure S4B in main Figure 4 and provide quantification of the co-localization of TBC1 and BIP to conclude on the ER localization of TBC1.

Figure 4B : Parasites were grown for 16h before auxin treatment, therefore they have undergone 2 cycles of division. Why a single parasite that appears as not having initiated its replication is shown ? The authors should show a representative image of the entire population.

Are the non-dividing parasites (one parasite / vacuole) found upon auxin treatment still alive ? The authors could address this point for instance by mechanically extracting those parasites from host cells and seed them on new cells.

Figure 4F-I :

The images presented in Figure 4 are not consistent leading to confusion in the interpretation of the data as also indicated in the text : « These IFA analyses clearly showed that parasites stained by IMC1 or MLC1 appeared to be morphologically intact (Figure 4F) in parasites grown for 16 hours but followed by auxin treatment for 13 hours. However, parasites were largely distorted with fragmented structures of the

IMC, and distorted distribution of the cytoskeleton (tubulin), as shown by staining of IMC1, GAP45 and Tubulin (Figure 4G-I). »

To appreciate the defects upon auxin treatment, the authors should allow the parasites to invade host cells for 6h and then treated with auxin for 16h. This will allow to clarify whether depletion of TBC1 leads to an early defect in daughter cell budding (at the first division cycle) correlated or not with impaired Golgi, centrosome, apicoplast duplication or just daughter bud formation (IMC/GAP45).

Please provide quantification for all these parameters (also for ER structure alteration) and clarify whether multinucleated cells are observed ?

In addition, the authors should provide electron microscopy (TEM) analysis of the mAID-TBC1 parasites treated with IAA to characterize in detail the morphological defects of the distinct parasite organelles, notably the ER, Golgi and mitochondrial in TBC1-mAID parasites treated with auxin.

In general, the data provided in Figure 4 argues for a role of TBC1 in the direct vesicular transport between the ER and the IMC (pellicle), a function not proposed by the authors in their conclusions ?

Figure S5 :

Similar to my previous comments, any conclusion on rhoptries and microneme integrity or localization can not be drawn with the used protocol. It may well be that the observed intact rhoptries, micronemes and dense granules originate from the mother cell. If a drastic defect in cytokinesis is demonstrated by the authors (after performing additional experiments), one may expect to see multinucleated parasites without formation of daughter buds and consequently no formation of apical mature rhoptries and micronemes in daughter cells. However, one will still observe mother cell apical organelles.

EM analysis should allow to address this important point.

The SORTLR staining appears as expected, not dispersed and apposed to the nucleus however the signal is too bright, saturated to make a firm conclusion on this aspect.

Also, GRA7 and GRA12 proteins seem to accumulate at the parasite plasma membrane and in the cytosol, displaying a secretion defect. If the parasites are grown for 16h before auxin treatment, a high amount of dense granule proteins will be already secreted in the vacuolar space when the parasites are observed by microscopy. Thus, the authors can not conclude anything on this point with the protocol they used. Again to appreciate the defects in secretory organelles upon auxin treatment, the authors should allow the parasites to invade host cells for 6h and then treated with auxin for 16h.

Fig 5G : What is the difference between type 5 and type 6 Golgi morphology ? It is not clear from the provided images.

Quantification indicates a very mild defect in Golgi morphology upon TBC1 depletion, which is characterized by a 13% appearance of type 6 Golgi morphology.

These findings indicate a minor role of TBC1 in the vesicular transport between the ER and the Golgi, in

opposite to what is claimed by the authors.

Please also quantify the percentage of parasites showing altered VAC and mitochondria morphology. What do the authors conclude from this result for the role of TBC1 in vesicular trafficking?

Figure 7

The TBC1 depleted parasites display very similar defects than previously observed for the Rab11A and Rab11B defective parasites, with a strong defect in cytokinesis.

Rab11A was the top pull-down partner with TBC1. Why the authors did not investigate this interaction ? Please perform similar experiments than the ones performed with Rab2 to assess a possible interaction between Rab11A/B and TBC1.

RESPONSE

Reviewers' comments:

Reviewer #1 (Remarks to the Author):

In their paper titled " Profiling of TBC domain-containing proteins reveals that the only essential protein TBC1 regulates the Early Vesicular Transport Pathway in Toxoplasma gondii," submitted on October 25, 2023, Sun et al investigate the importance of TBC domain proteins in the apicomplexan parasite Toxoplasma gondii. TBC domain proteins are known to be Rab GTPase regulators in eukaryotes. Rabs have been demonstrated to play a crucial role in the Toxoplasma lytic cycle. Studying how these Rabs are regulated is particularly interesting, especially given that very little was known about them.

Using Cas9 to tag and then disrupt the genes (straight KO, then Auxin for TBC1, which could not be obtained directly), the authors examined 17 TBC proteins identified using ToxoDB and bioinformatics analysis. Focusing on the most essential protein, TBC1, they conducted a comprehensive investigation of the impact of the gene's depletion using the auxin degradation system. They notably deeply investigate the state of the parasite's organelles. They also carried out point mutations on the dual-finger part of the TBC domain.

The results showed that TBC1 is crucial for the trafficking of vesicles from the endoplasmic reticulum (ER) to the Golgi apparatus. Finally, using pulldown assays, they demonstrated that TBC1's primary partner is Rab2, and that TBC1 is also essential in Type 2 strains.

Response: We greatly appreciate the positive comments from the reviewer. We have a point-by-point response to the detailed comments and suggestions.

Major issue:

The paper is well-structured, with well-presented results that require only minor modifications for clarity. However, there is a major ethical issue that needs attention.

As mentioned, the paper was submitted on October 25, 2023. Since May 29, 2023, a paper titled "Toxoplasma gondii encodes an array of TBC-domain containing proteins, including an essential regulator that targets Rab2 in the secretory pathway," authored by Quan et al, has been available on BioRxiv

(<https://www.biorxiv.org/content/10.1101/2023.05.28.542599v1.full>).

The striking homology between these two papers, from the technics used to the organization of the content, is hard to overlook. On ToxoDB, all the TBC proteins mentioned in the BioRxiv paper were already assigned gene IDs in user comments since May 29. TBC1 in the submitted paper is identified as TBC9 in the BioRxiv paper.

The lack of any reference to the BioRxiv paper in the submitted manuscript is concerning. At a minimum, a citation to the BioRxiv paper should have been included.

Have the authors of the BioRxiv paper have been contacted regarding this overlap? If the similarity between the two papers is purely coincidental, a back-to-back publication strategy would have been preferable to uphold good scientific practice, especially given the relatively small field of Toxoplasma research. The BioRxiv paper should already be under revision somewhere.

This matter need clarification, because in it state I can not approve any publication.

Response: Great thanks to the reviewer for the comments and suggestions. In recent years, we have focused on the mechanisms of endocytosis, vesicular transport and nutrient acquisition (see publications, PMID: 36813769, 37108334, 37548452 and 37941035), which has led us to dissect the regulation of endocytic trafficking and vesicular transport in the parasite. In 2019, one of my PhD students started this project of TBC proteins. She has intensively worked on it for four years. We assure that it is purely coincidental regarding the conception and design of the project. In addition, we have been working on additional experiments after submission of the past version of manuscript. These experiments include the impact of TBC9 on IMC formation and organelle replication, electron microscopy of parasites depleted with TBC9, and the interaction of TBC9 with Rab11A and Rab11B (see additional results in Figure 6, 7, and 9). We noticed the preprint before the submission, but did not try to contact the authors. In the current version of manuscript, we have added and discussed the main findings from the preprint reference. We have also changed the names of these TBC domain-containing proteins in the current version of manuscript, to make the nomenclature consistent with the TOXODB information (updated by the preprint authors), and to make the two papers easier to understood. We hope that we have clarified our work and situation, and we kindly request the reviewer's endorsement for the publication of this work.

Minor revisions:

After addressing this disturbing issue, I will proceed with my comments on the article

Experiments:

In the results section, the last paragraph discusses the evaluation of the impact of TBC1 knockout in type 2, even though the authors have already clearly demonstrated its essential role in type 1. It would be more interesting to assess bradyzoite differentiation in type 2 rather than solely examining the KO's impact. If no additional analysis is planned for type 2, it might be sufficient to mention this aspect briefly when discussing the generation of their mAID strain in the section titled 'TBC1 is essential for parasite replication,' rather than dedicating a whole separate section to it

Response: Great thanks for the suggestions. We agree that the ME49 type line does not provide additional information unless the bradyzoite differentiation was significant. Considering there was no actual effect of the protein on the bradyzoite differentiation, we have moved the content and provided just a brief description of the ME49 type line in the generation of the mAID strain, as suggested.

Text revisions:

As mentioned earlier, TBC1 corresponds to TBC9 in the BioRxiv paper. Since ToxoDB has already been annotated with the 18 TBC proteins reported in the BioRxiv since May 2023, it would be advisable to standardize the nomenclature to avoid confusion in the future.

Response: Great thanks to the suggestions. We have revised the names of TBC domain-containing proteins identified in the study, according to the information in the TOXODB, in the current version of the manuscript.

On line 199, the conclusion that 'most of the TBC proteins are localized in cellular structures associated with protein secretion or protein sorting pathways' appears to be somewhat overstated. While 8 out of 17 are indeed located in the ER/Golgi area, 9 are distributed in the cytoplasm, residual body, or remain unlocalized.

Response: Great thanks to the comments. We have revised the sentence to “nearly half of the TBC proteins....”.

On lines 244-256, the explanation of the observed phenotype for the Inner Membrane Complex (IMC) may benefit from clarification. The author could clarify that, in 87% of cases, no impact is observed on the IMC, but a small proportion of parasites exhibit IMC distribution issues, accounting for 13%.

Response: Great thanks to the comments. In the present manuscript, we conducted a detailed analysis of the phenotype associated with disrupted IMC (Figure 6). Parasites were induced at different time points, and based on the condition of the IMC and the nuclei, we categorized the parasites into four types and performed quantitative analysis. The results demonstrated that depletion of TBC9 affected the biogenesis and maturation of daughter IMC. Rab11A controls IMC assembly, whereas Rab11B is essential for IMC biogenesis in daughter parasites. Surprisingly, we observed protein interactions between TBC9 and Rab11A, as well as Rab11B, providing robust support for the regulation of IMC biogenesis and maturation by TBC9.

On line 305, I am wondering about whether the parasites with diffuse Golgi correspond to those with diffuse IMC1. Since both are observed in the same proportion (13%), it could be interesting to perform a dual-label IMC1-Golgi.

Response: Great thanks to the comments. We concur with your viewpoint. In fact, in our initial manuscript submission, we performed dual staining of the parasite IMC1 and the Golgi. To enhance clarity, we removed redundant layers from Figure 5C. The results revealed that parasites with a dispersed Golgi did not correspond to parasites with disrupted IMC1 structure, suggesting that any similarity in the data was coincidental. Additionally, our induction of TBC9 prior to the first round of replication showed that under these induction conditions, over 12% of parasites exhibited a diffused Golgi (Figure 6I-J), while only approximately 5% displayed disrupted IMC1 (Figure 6G-H), further supporting the limited association between the two phenotypes.

Lastly, on line 592, there seems to be a reference error (Error! Bookmark not defined).

Response: Thank you for pointing out the issue. We have rechecked the references and ensured that they display correctly after converting to PDF format.

Figure Comments:

In general, there are some key quantifications that are missing and should be included in the figures. These could include data on the proportion of parasites with disrupted organelles due to TBC1 KO, as well as more detailed information on fine co-localization.

Response: Great thanks to the suggestions. We have quantified the proportion of various organelle impairments caused by TBC9 (formerly TBC1) knockout in parasites, including the ER, SORTLR, VPS35, CPL, and HSP60 (Figure 5). We also measured the Pearson correlation coefficient (PCC) between the ER-localized TBC1, 5, 6, 9, 14, and the ER marker BIP to provide more detailed co-localization information (Figure 2).

Regarding Figure 2, the co-localization figures in Figure S2 appear to be more informative than those presented in Figure 2. It may be beneficial to consider swapping them, for the protein with a well-defined localization.

Response: Great thanks to the suggestion. We have swapped the co-localization figures in Figure S2 to those presented in Figure 2, as suggested.

For Figure 3, there seems to be a small issue with the representative pictures for TBC9 and TBC12, as they don't align with their respective area analysis. In the picture, TBC9 plaques appear larger than TBC12, whereas the area analysis suggests otherwise.

Response: Great thanks to the suggestion. We have used a more representative plaque picture of TBC3 (formerly TBC12), which is consistent with the plaque size analysis, that is, TBC15 (formerly TBC9) > TBC3 (formerly TBC12) (Figure 3).

In Figure 4, the TBC1 m-AID plaque without IAA appears smaller than the Tir1 control. It would be helpful to provide some commentary or explanation for this observation.

Response: Great thanks to the suggestion. We have employed more representative plaque pictures of Tir1 and TBC9-mAID lines to provide more precise information (Figure 4D).

Figure 5: It might be a good idea to group all the IFA data from both Figure 5 and Figure S5 together. This could make it easier for readers to follow and compare the results.

Response: Great thanks to the suggestion. We have grouped all the IFA data from both Figure 5 and Figure S5 as much as possible to make it easier for readers to follow and compare the results (Figure 5).

For Figure 6, adding some quantification analysis could enhance the understanding of the impact of the different point mutations.

Response: Great thanks to the suggestion. We have conducted additional replication experiments and performed quantitative analysis on the plaques displayed in Figure 8G (Figure 8F and L). The data revealed that the TBC9^{R74A} mutant exhibited smaller plaque sizes compared to the TBC9^{Q101A} mutant in the absence of IAA induction. This observation

suggests that the arginine residues in TBC9 may play a more critical role in protein activity, similar to mammalian TBC proteins.

In Figure 7, the Co-IP figure appears somewhat complex. Creating a small schematic to indicate that Rab 2 and 18 are tagged with Mbp, TBC1 with His, and that different pull-down conditions are applied to both the wild-type (WT) and point mutation samples could significantly improve reader comprehension.

Response: Great thanks to the suggestion. We have created schematic of pull-down assay (Figure 9C). The proteins mbp-Rab, mbp-RabQL (GTP-locked mutants), TBC9, and TBC9^{R74A} (GAP-inactive mutants) were expressed in *E. coli* and purified respectively. The mbp-Rab or mbp-RabQL proteins were used as bait and incubated with MBP agarose resin in tubes. After washing, the resin bound with mbp-Rab or mbp-RabQL was individually incubated with TBC9 or TBC9^{R74A}. Subsequently, the samples were thoroughly washed and subjected to Western blot analysis.

Reviewer #2 (Remarks to the Author):

Ming Sun et al explore the function of the TBC family of Rab GTPase Activating Proteins in *Toxoplasma tachyzoites*. By first screening the 17 member TBC family for expression, localization, and essentiality, they were able to focus on a single essential TBC protein called TBC1. They determined that TBC1 is an ER protein that is critical for regulating protein sorting from the ER to the Golgi. Loss of TBC1 resulted in morphological defects, organellar defects, and lethality due to aberrant replication. Furthermore, the authors determined that TBC1 requires a dual finger catalytic site and that may interact Rab proteins such as Rab2, as expected for a TBC protein but not greatly supported by their data. Altogether, the authors resolve some of the functional redundancy within an important gene family (TBC) and provide some mechanistic insights into the essential role of TBC1 in tachyzoites. This study is novel and opens the door for further investigation of Rab-dependent processes in *Toxoplasma* that could be targeted by drugs to treat *Toxoplasma* infections.

Response: We greatly appreciate the positive comments from the reviewer. We have a point-by-point response to the detailed comments and suggestions.

Major concerns

1. There seems to be very little agreement between the TBC knockout phenotypes generated here (Fig. 3) and the CRISPR/Cas9 KO screen fitness scores reported by Sidik et al. 2016 ref 6. Specifically referring to TBC2,3,4,5,9,10,12,13,16. Meaning those with low CRISPR fitness scores formed larger plaques than those with higher CRISPR fitness scores, opposite of expectations. There is so much inconsistency that it casts some doubt on the identities of the mutants shown. Please double check that the data and representative images perfectly match the mutants. Since genome-wide CRISPR screens have false positives and false negatives due to inefficiencies and off-targets, there may be a reasonable explanation for the

inconsistencies. Please dedicate a paragraph in the discussion to explain this discrepancy.

Response: Great thanks to the reviewer for the positive comments. We have carefully examined whether the data and representative images match the mutants completely. Although we have replaced the representative images of some genes (TBC1, formerly known as TBC10, and TBC3, formerly known as TBC12), and used a more common number of pixels to indicate the size of the plaques, the overall phenotype of the TBC knockout lines is generally consistent with the previous manuscript, showing significant differences with fitness scores. We provide an explanation for this phenomenon in the discussion section, “This discrepancy may be attributed to the relatively low efficiency and potential off-target effects of genome-wide CRISPR screening, leading to false positives and false negatives. Additionally, the scoring accuracy for non-essential genes (those that can be deleted) may be more prone to error in genome-wide CRISPR screens”, as suggested.

2. The TBC1-6HA co-IP data shown in Figure 7B is difficult to interpret because the parental negative control data is not included in Supplement Dataset 1. There should be 2 replicates for control, 2 replicates for the test, averaged data with fold change and P values. The link between TBC1 and Rab2 is extremely spurious based on the co-IP data (I believe it ranks 414 based on fold enrichment). A better presentation of the data file and more accurate description of the results is needed. Any additional method to show TBC1 interaction with Rabs by TurboID or reciprocal co-IP with western blotting would be more convincing than the recombinant protein data shown in Fig. 7C.

Response: Great thanks to the reviewer for the positive comments. Actually, our dataset includes parental negative control data. To enhance their visibility, we eliminated surplus data and simplified the nomenclature of the parental negative control data as RH-1 Area and RH-2 Area. Additionally, we incorporated P values into the dataset, as suggested. In fact, we performed TurboID experiments. However, due to the broad cellular localization of TBC9 (cytoplasmic regions associated with the ER, formerly TBC1) and the transient interaction between TBC proteins and Rab proteins inside cells, the proteomic results had limited reference value (with numerous nonspecific proteins and minimal to no detection of Rabs). Furthermore, following the acquisition of the proteomics results through CO-IP, we initially attempted to validate the interactions directly using the CO-IP method. However, due to the transient nature of their interactions in the cell, we were unable to effectively pull down the interacting Rab proteins in amounts sufficient for detection by western blot. Therefore, after an extensive review of the literature, we resorted to in vitro validation of interactions between the mutant forms, which allowed for more stable binding to facilitate detection.

Minor concerns

1. Title: The title is misleading. Other TBC proteins may also regulate early vesicular transport. Consider revising to "Profiling of TBC domain-containing proteins reveals TBC1 as an essential regulator of the early vesicular transport pathway in *Toxoplasma gondii*" or further clarify the original title.

Response: Thank you for the suggestion. We have revised the title to “The only essential TBC-domain protein TBC9 regulates early vesicular transport pathway and IMC formation in *Toxoplasma gondii*”, as requested by the journal to present the title in less than 15 words.

2. Line 21: write out TBC in abstract.

Response: Thank you for the suggestion. We have revised the sentence to “ ..., TBC (Treg2/Bub2/Cdc16) domain-containing proteins.”

3. Line 23: Functional profiling of these proteins in tachyzoites...

Response: Thank you for the suggestion. We have revised the sentence to “Functional profiling of these proteins in tachyzoites...”

4. Line 41: Delete inciting

Response: Thank you for the suggestion. We have deleted “inciting” and revised the sentence to “the tachyzoites that is responsible for acute infection, and the bradyzoites (in tissue cysts) that cause a chronic infection”.

5. Line 43: Delete become established and

Response: Thank you for the suggestion. We have deleted “become established and” and revised the sentence to “Although most of *T. gondii* infections are asymptomatic, latent infections in the brain and muscle tissues can lead to severe symptoms in immunocompromised patients.”

6. Line 48: Change operations to manipulation

Response: Thank you for the suggestion. We have changed “operations” to “manipulation” and revised the sentence to “More importantly, *T. gondii* can serve as a model organism for studying members of the Apicomplexa phylum, due to its ease of in vitro culturing and genetic manipulation.”

7. Line 133: It would be simpler to just say you identified 17 TBC proteins by searching ToxoDB using the Interpro ID IPRO35969 Rab-GAP-TBC domain superfamily.

Response: Thank you for the suggestion. We used the "IPRO35969 Rab-GAP-TBC domain superfamily" to search ToxoDB, filtered by organisms. This search yielded 5,839 results, including many proteins related to Rab. Conversely, as the past version of manuscript, when performing a "TBC" search on ToxoDB and filtering by organism, we identified only 22 results, which could be readily screened using Interpro.

8. Line 212: Period after BIP.

Response: Thank you for the suggestion. We have checked and revised punctuation marks.

9. Lines 245-251: Please better connect the treatment conditions and controls with the observed phenotypes.

Response: Great thanks to the suggestion. In the present manuscript, we conducted a detailed analysis of the phenotype associated with disrupted IMC (Figure 6). Parasites were induced at

different time points, and based on the condition of the IMC and the nuclei, we categorized the parasites into four types and performed quantitative analysis. The results demonstrated that depletion of TBC9 (formerly TBC1) affected the biogenesis and maturation of daughter IMC. Rab11A controls IMC assembly, whereas Rab11B is essential for IMC biogenesis in daughter parasites. Surprisingly, we observed protein interactions between TBC9 and Rab11A, as well as Rab11B, providing robust support for the regulation of IMC biogenesis and maturation by TBC9.

10. Figure 1: How were the TBC genes assigned numbers 1-17?

Response: Thank you for the comments. We have revised the names of TBC domain-containing proteins identified in the study, according to the information in the TOXODB, in the current version of the manuscript.

11. Figure 2: Do the TBC protein localizations match the Hyper-LOPIT study? Barylyuk et al. 2020 CHM.

Response: Thank you for the comment. As you suggested, after comparing our experimental results with the predicted results of Hyper-LOPIT, it was found that 7 of the 17 TBC proteins in this study were not predicted by Hyper-LOPIT or the predicted results were unknown, and the location of the other 10 proteins was also inconsistent with the Hyper-LOPIT study.

12. Figure 3B: It is unclear what Plaque Area % means. For instance, what does 7% plaque area mean for RH control? Should % simply mean mm^2 or pixels^2 . Please explain why unpaired T-test and Welch's T-test were used and which samples they were used for.

Response: Thank you for the comments. In the current manuscript, we have used a more commonly accepted number of pixels to indicate the size of the plaque. In addition, the unpaired t-test and Welch's t-test were performed on data with a normal distribution, with or without homogeneity of variance, respectively. However, due to updates in the data, only the unpaired t-test was utilized for the statistical analysis in the current manuscript.

13. Figure 4B/Lines 229/230: It appears the mAID-3HA tag on TBC1 was responsible for an innate hypomorphism (smaller plaques). This should be acknowledged in the results at lines 229/230.

Response: Thank you for the suggestion. We have replaced a more representative picture as figure 4B.

14. Figure 4F-I: Label panels that are considered normal (F-G) or fragmented (H-I) in the figure.

Response: Thank you for the suggestion. We have added label panels in the figure 6A-D to indicate normal IMC or fragmented IMC.

Reviewer #3 (Remarks to the Author):

In this study, Sun et al. examined the role of the TBC domain containing protein family in

Toxoplasma gondii. By bioinformatic search, they identified 17 genes encoding TBC proteins, among which only TBC1 (out of 17 identified TBC proteins) was found to be essential for the life cycle of the parasite in fibroblasts. Based on localization study after endogenous tagging of the protein, the authors concluded that TBC1 is localized in the ER. Inducible knock-down approach showed that TBC1 is essential for parasite replication. Examination of different parasite organelles by immunofluorescence showed that the pellicule (tubulin, actin cytoskeleton and IMC), the Golgi, the mitochondria, the ER and the VAC (but not the rhoptries and micronemes) morphology is altered upon TBC1 depletion. The authors concluded that TBC1 regulates trafficking between the ER and the Golgi.

The study is interesting for the *Toxoplasma* community providing new knowledge on the regulation of vesicular trafficking in this parasite via the identification and description of novel GAPs. However, while the role of TBC1 in parasite replication through its conserved dual-finger active sites (point mutation approach) has been clearly demonstrated, major conclusions can not be drawn from the results provided in this study :

Response: We greatly appreciate the positive comments from the reviewer. We have a point-by-point response to the detailed comments and suggestions.

1) The localization of TBC1 in the ER can not be concluded from the provided images, notably due to a lack of resolution and saturated signals. A major part of the TBC1 signal does not co-localize with the ER.

Response: Great thanks to the comments. For their higher specificity and ability to provide non-saturated signals, we employed anti-BIP antibodies to investigate the colocalization of these TBC proteins with the ER and provide images zoomed in the ER region (Figure 2A and S2D). We calculated the Pearson's correlation coefficient (PCC) between these TBC proteins and BIP to provide more detailed information on their colocalization. In fact, some reported ER proteins do not exhibit typical reticular localization around the nucleus, contributing to the variation in subcellular localization observed among various ER markers, including the differential localization of BIP and cis-IPPS. The average PCC between TBC9 (formerly TBC1) and BIP is 0.73, indicating a significant co-localization of TBC9 with the major components of the endoplasmic reticulum (ER). Consequently, we characterize TBC9 as being associated with the endoplasmic reticulum and its related regions.

2) The role of TBC1 in regulating vesicular transport between the ER and the Golgi can not be concluded due to pleiotropic effects on most of the parasite organelles, lack of clear TBC1 localization in the ER, only very partial effect on the Golgi apparatus (not explained) and what seems to be a very drastic defect in early phase of daughter pellicle formation. Loss of parasite polarity in absence of daughter bud formation leads to pleiotropic effects on most of the parasite organelles.

Response: Great thanks to the comments. Depletion of TBC9 (formerly TBC1) during daughter bud formation indeed leads to the loss of parasite polarity, as demonstrated by our electron microscopy data (Figure 7I). However, quantitative analysis of the ER phenotype following TBC9 depletion showed that over 80% of parasites exhibited disrupted ER structures (Figure 5B), which significantly exceeded the 13% of parasites showing polarity

loss due to the absence of daughter buds (Figure 6E). This suggests a limited association between the severe ER defects and the pleiotropic effects caused by polarity loss. Furthermore, we conducted a more detailed quantitative analysis of Golgi defects (Figure 5D). The results revealed a significant increase in fragmented Golgi and a noticeable decrease in individual Golgi units in the none daughter bud stage, indicating the impact of TBC9 depletion on the Golgi is also minimally associated with polarity loss-induced pleiotropic effects.

Moreover, we conducted a more comprehensive analysis of TBC9 localization, which confirmed its distribution in the ER. Importantly, we demonstrated the interaction between TBC9 and Rab2, which plays a role in regulating vesicular transport between the ER and the Golgi. Based on these findings, we propose that TBC9 plays a role in regulating vesicular transport between the ER and the Golgi.

3) The authors partly based their conclusions on the interaction between Rab2 and TBC1, although Rab11A was the top pulled-down partner of TBC1 and indeed, the TBC1 depleted parasites share many defects observed in Rab11A/B defectives parasites, previously described in the literature.

Response: We greatly appreciate your insightful feedback. Based on pull-down experimental evidence, we have successfully demonstrated the interaction between TBC9 and both Rab11A and Rab11B (Figure 9D). This interaction serves as robust evidence supporting the crucial role of TBC9 in both the biogenesis and maturation processes of the daughter IMC.

Specific comments:

Figure 2: How do the authors explain the localization of some TBC in the residual body ?

Response: Great thanks to the comments. In the discussion section, we have supplemented an elucidation of the subcellular localization of TBC10 within the residual body. The residual body encompasses the ER, mitochondria, and plasma membrane left by the dividing parent, as well as the dense granular proteins secreted by parasites and the acidocalcisome that accumulates during division, all enveloped by the intravacuolar network (IVN). The localization of the TBC10 in the RB suggests that they may be retained by the parent or contribute to other functions, such as potential involvement in membrane transport between progeny tachyzoites. Moreover, in the previous manuscript, TBC16 (formerly TBC17) and TBC14 (formerly TBC16) were also found to be distributed within the residual body. However, after referring to a biorxiv manuscript with a similar topic, we conducted additional IFA experiments on these TBC proteins. We found that TBC16 primarily localizes to the cytoplasm, while TBC14 predominantly localizes to the ER. The residual body localization of TBC14 is attributed to the retention of ER components in the residual body during parasite division.

Line 165 : Immunofluorescence assays (IFA) showed that the TBC1, 3, 10, 13 and 16 were primarily located in the cytoplasmic region or the endoplasmic reticulum (the ER) (Figure 2A). Colocalization studies with a ER marker cis-IPPS (decaprenyl diphosphate synthase) (43) showed at least a partial co-localization with the ER was observed for these TBC

proteins (Figure S2D).

Do the authors consider that all these cited TBC localize in the cytosol « or » the ER, or rather that all these TBC localized in the cytosol AND the ER ?

Response: Great thanks to the comments. We consider all these cited TBC localize in the cytosol and the ER and we have revised the sentence to "...located in the cytoplasmic region and the endoplasmic reticulum (the ER)".

In Fig S2, please provide better quality images including improved resolution, zoom in the ER region as well as a non-saturated signal for the ER marker cis-IPPS.

Response: Great thanks to the suggestions. For their higher specificity and ability to provide non-saturated signals, we employed anti-BIP antibodies to investigate the colocalization of these TBC proteins with the ER and provide images zoomed in the ER region (Figure 2A and S2D).

Figure 3 : TBC10 seems to be also essential. How do the authors successfully maintain those parasites in culture ? the plaque area suggests that they do not replicate.

Response: Great thanks to the comments. TBC1 (formerly TBC10) was successfully knocked out, although the knockout line only formed extremely small plaques. Notably, the growth of the TBC10 knockout line was slower compared to other knockout strains, requiring a higher inoculum of parasites in each passage to maintain their growth during cultivation.

Figure 4 : Please place the results of figure S4B in main Figure 4 and provide quantification of the co-localization of TBC1 and BIP to conclude on the ER localization of TBC1.

Response: Great thanks to the suggestion. We included Figure S4B in the main Figure 2 for a comprehensive analysis of ER-localized TBC proteins. Additionally, we calculated the Pearson's correlation coefficient (PCC) between these TBC proteins and BIP to provide more detailed information on their colocalization (Figure 2B).

Figure 4B : Parasites were grown for 16h before auxin treatment, therefore they have undergone 2 cycles of division. Why a single parasite that appears as not having initiated its replication is shown ? The authors should show a representative image of the entire population.

Response: Thank you very much for pointing out the issue. This is a descriptive error in the figure legend, and we have revised the figure legend to "TBC9-mAID-3xHA parasite line was grown in IAA or ethanol for 24 hours, followed by fixation for IFA analyses"

Are the non-dividing parasites (one parasite / vacuole) found upon auxin treatment still alive ? The authors could address this point for instance by mechanically extracting those parasites from host cells and seed them on new cells.

Response: Great thanks to the suggestion. We found that the non-dividing parasites (one parasite / vacuole) observed after auxin treatment were already dead when we mechanically extracted them from host cells and seeded them onto new cells, as suggested.

Figure 4F-I :

The images presented in Figure 4 are not consistent leading to confusion in the interpretation of the data as also indicated in the text : « These IFA analyses clearly showed that parasites stained by IMC1 or MLC1 appeared to be morphologically intact (Figure 4F) in parasites grown for 16 hours but followed by auxin treatment for 13 hours. However, parasites were largely distorted with fragmented structures of the IMC, and distorted distribution of the cytoskeleton (tubulin), as shown by staining of IMC1, GAP45 and Tubulin (Figure 4G-I). »

To appreciate the defects upon auxin treatment, the authors should allow the parasites to invade host cells for 6h and then treated with auxin for 16h. This will allow to clarify whether depletion of TBC1 leads to an early defect in daughter cell budding (at the first division cycle) correlated or not with impaired Golgi, centrosome, apicoplast duplication or just daughter bud formation (IMC/GAP45).

Please provide quantification for all these parameters (also for ER structure alteration) and clarify whether multinucleated cells are observed ?

Response: Great thanks to the comments and suggestions. Under the induction condition you suggested, we observed and categorized phenotypes of IMC1, GAP45, and nuclear into four types. We found that depletion of TBC9 still resulted in IMC disruption in budding parasites (type D) under this induction condition (Figure 6G). However, the proportion of such parasites was less than 5%, significantly lower than the 13% observed in parasites that underwent 1-2 rounds of division before induction (Figure 6E and G). Additionally, there was a significant increase in the number of parasites in the post-nuclear division stage but lacking daughter buds (type B). These data indicate that depletion of TBC9 hinders de novo synthesis and assembly of the daughter IMC, leading to difficulties in daughter bud formation even after nuclear division has occurred.

Following your suggestion, we also examined whether the duplication of the Golgi, centrosomes, and apicoplast was affected under this induction condition. Quantitative analysis revealed a significant increase in the number of unreplicated Golgi, centrosomes, and apical complexes, indicating compromised replication of these organelles.

Furthermore, we did observe multinucleated cells (types B/C/D), although the majority of these cells only had two nuclei, possibly due to premature parasite death following TBC9 depletion. Certainly, as per your request, we conducted observations and quantification of the ER under this induction condition. However, due to the similar phenotypic and quantitative results obtained by first growing the parasites for 16 hours and then inducing them for 13 hours, we have presented the results from the 16+13 induction condition (Figure 5A and B) for the sake of conciseness and clarity in the manuscript. In summary, these data demonstrate that depletion of TBC9 leads to early defects in daughter cell budding, as well as impaired replication of the Golgi, centrosomes, and apicoplast.

In addition, the authors should provide electron microscopy (TEM) analysis of the mAID-TBC1 parasites treated with IAA to characterize in detail the morphological defects of the distinct parasite organelles, notably the ER, Golgi and mitochondrial in TBC1-mAID parasites treated with auxin.

Response: Great thanks to the comments. We conducted electron microscopy analysis of TBC9-depleted parasites, which revealed clear evidence of abnormal folding and disrupted

network organization in the ER (Figure 7B-C1). We also observed varying degrees of folded mitochondria (Figure 7C2-D), as well as loosely dispersed vesicle-like Golgi (Figure 7E-E1).

In general, the data provided in Figure 4 argues for a role of TBC1 in the direct vesicular transport between the ER and the IMC (pellicle), a function not proposed by the authors in their conclusions ?

Response: Great thanks to the comments. Depletion of TBC9 indeed has a significant impact on both the ER and the IMC. However, based on our more detailed quantitative analysis of Golgi defects, as previously mentioned, the dispersed phenotype of the Golgi is not solely due to IMC disruption. In fact, following TBC9 depletion, the degree of Golgi dispersion and fragmentation intensifies regardless of whether daughter buds are formed (Figure 5D).

Therefore, we cannot overlook the role of TBC9 in the Golgi. Furthermore, our pull-down experiments provide evidence of the interaction between TBC9 and Rab2, further supporting the involvement of TBC9 in vesicular transport between the ER and the Golgi.

Figure S5 :

Similar to my previous comments, any conclusion on rhoptries and microneme integrity or localization can not be drawn with the used protocol. It may well be that the observed intact rhoptries, micronemes and dense granules originate from the mother cell. If a drastic defect in cytokinesis is demonstrated by the authors (after performing additional experiments), one may expect to see multinucleated parasites without formation of daughter buds and consequently no formation of apical mature rhoptries and micronemes in daughter cells.

However, one will still observe mother cell apical organelles.

EM analysis should allow to address this important point.

Response: Great thanks to the comments and suggestions. The previous induction method was designed to investigate whether the secretion of secretory organelles in normally shaped parasites would be affected by TBC9 depletion. We allowed parasites to invade host cells for 6 hours and subsequently treated them with auxin for 16 hours to observe the formation of rhoptries and microneme. Our electron microscopy analysis revealed that the mother parasites exhibited intact rhoptries and microneme. However, the daughter parasites only displayed early components of the cytoskeleton recruited during the onset of budding and did not exhibit mature rhoptries and microneme, as anticipated (Figure 7H).

The SORTLR staining appears as expected, not dispersed and apposed to the nucleus however the signal is too bright, saturated to make a firm conclusion on this aspect.

Response: Thank you for pointing out this issue. We have replaced the overexposed images with clearer ones to better visualize the phenotype. Approximately 80% of the parasites exhibited a certain degree of dispersion in SORTLR localization after TBC9 depletion. However, the main fraction of SORTLR still localizes to the perinuclear region, which could potentially explain the unaffected functionality of SORTLR.

Also, GRA7 and GRA12 proteins seem to accumulate at the parasite plasma membrane and in the cytosol, displaying a secretion defect. If the parasites are grown for 16h before auxin treatment, a high amount of dense granule proteins will be already secreted in the vacuolar

space when the parasites are observed by microscopy. Thus, the authors can not conclude anything on this point with the protocol they used. Again to appreciate the defects in secretory organelles upon auxin treatment, the authors should allow the parasites to invade host cells for 6h and then treated with auxin for 16h.

Response: Great thanks to the comments and suggestions. We have re-examined the secretion of GRA7 and GRA12 as suggested (Figure 5I). We discovered that GRA7 and GRA12 can still be properly secreted into the parasitophorous vacuole even after TBC9 depletion. Furthermore, electron microscopy analysis also showed no significant accumulation of dense granules in TBC9-depleted parasites (Figure 7). It has been reported that overexpression of dominant-negative Rab11A leads to defects in dense granule secretion. Our phenotype and interaction results suggest that TBC9 may regulate Rab11A, therefore, the lack of dense granule secretion defects in TBC9 knockdown parasites is contrary to our expectations. One possible explanation is that the transport of dense granules, mediated by Rab11A, and the assembly of the IMC are regulated by different TBC proteins, which warrants further investigation.

Fig 5G : What is the difference between type 5 and type 6 Golgi morphology ? It is not clear from the provided images.

Quantification indicates a very mild defect in Golgi morphology upon TBC1 depletion, which is characterized by a 13% appearance of type 6 Golgi morphology.

These findings indicate a minor role of TBC1 in the vesicular transport between the ER and the Golgi, in opposite to what is claimed by the authors.

Response: Great thanks to the comments and suggestions. To enhance image clarity, redundant layers have been removed from the images (Figure 5C). While type 5 Golgi appeared fragmented, it could still be observed as individual entities. However, type 6 Golgi exhibited severe dispersal, making their independent identification impossible. As previously described, Golgi in different states (during daughter formation or in the absence of daughter parasites) exhibited an increased tendency for fragmentation and dispersal. Ultimately, Golgi undergoing division during daughter budding suffered the most severe damage, with over 25% of parasites exhibiting the most pronounced phenotype (type 6) (Figure 5D). This evidence supports the crucial role of TBC9 in the vesicular transport between the ER and the Golgi.

Please also quantify the percentage of parasites showing altered VAC and mitochondria morphology. What do the authors conclude from this result for the role of TBC1 in vesicular trafficking?

Response: Great thanks to the comments and suggestions. In accordance with your request, a quantitative analysis was conducted on parasites exhibiting morphological alterations in VAC and mitochondria. The obtained data demonstrated that approximately 50% of the parasites exhibited an increased extent of VAC dispersion, while ~ 90% of the parasites displayed morphological abnormalities in their mitochondria (Figure 5K). Vacuoles (VAC) are known to localize to the late secretory pathway, and SORTLR and VPS35, which also exhibit the dispersal phenotype, are all localized in this pathway. This suggests that TBC9 may be involved in vesicular transport in this region. Previous studies have demonstrated that some

Rab GTPases located to this region, including Rab7, Rab5A, Rab5B, and Rab5C. Therefore, investigating the interaction between TBC9 and these Rab GTPases can provide insights into the regulatory role of TBC9 in the late secretory pathway. For *T. gondii* mitochondria, the majority of mitochondrial proteins are encoded by the nucleus and must be targeted and transported to the mitochondria after translation. This process may involve vesicular transport mediated by TBC9. Unfortunately, there are currently no reports on the involvement of any Rab GTPases in vesicular transport processes related to mitochondria. Therefore, further investigation is required to elucidate the role of TBC9 in this process.

Figure 7

The TBC1 depleted parasites display very similar defects than previously observed for the Rab11A and Rab11B defective parasites, with a strong defect in cytokinesis.

Rab11A was the top pull-down partner with TBC1. Why the authors did not investigate this interaction ?

Please perform similar experiments than the ones performed with Rab2 to assess a possible interaction between Rab11A/B and TBC1.

Response: Great thanks to the comments and suggestions. As previously mentioned, we conducted pull-down experiments in accordance with your suggestion, which confirmed the interaction between Rab11A/B and TBC1. This finding provides support for the role of TBC9 in the formation of the IMC.

Reviewers' comments:

Reviewer #1 (Remarks to the Author):

First, I would like to thank the author for addressing my concerns and responding to all my comments. A significant improvement has been made to the paper, making it easier to follow and more convincing. However, there are still some minor points that need to be addressed:

Minor points:

- Figure 2B: Please specify "Pearson's correlation with BIP" instead of just "Pearson's correlation" for easier understanding of the graphic.
- Line 180 / Figure 2: The authors mention the potential involvement of the TBC in vesicular transport frequently. For the cytoplasmic TBC, were any colocalization studies with the ELC performed using markers like pro-M2AP?
- Line 423 / Figure 8: While the authors clearly demonstrate that complementation with TBC1 wt has no effect on the parasites and different markers, and the effects of the mutant TBC1 R74A and TBC1 Q101A are also clearly demonstrated in the replication and plaque assays, there is no information on the impact of mutations on the organelles previously shown to be strongly affected by TBC1 depletion (ER, Golgi, and IMC1). Are these organelles impacted in the same way?
- Figure 9D: The Western blot is somewhat confusing. Is the Input truly the input, corresponding to the protein before loading on the column? Does Bound correspond to elution? Was the wash checked to ensure no more unbound TBC9 could be detected? If the input is indeed the input, how does such a small amount of Mbp-Rab11B Q71L lead to such a strong signal in the bound fraction in the last well with His-TBC9 R74A?
- Line 720: Some typos or indications of corrections are still present in the text

Suggestions:

In this section I will provide some remarks for the authors to consider. The authors are free to follow them or not, as everybody as their own presentation and writing preference. These suggestions do not significantly impact the paper but could enhance understanding:

- The term "essential" is used frequently, but it should be clarified that the gene is "essential" only for the lytic cycle. As mentioned in the rebuttal letter, TBC9 KD does not affect bradyzoite differentiation and is therefore non-essential for this process and perhaps for the cat's life cycle. It should be specified that the essential aspect pertains to the lytic cycle
- Line 37: "Facultative life cycle" sounds somewhat odd. Consider using "asexual" or "lytic cycle" instead.
- Figure 2: Removing all the "6-TY" indications in the figure labeling might improve readability.
- Figures 5/6: These figures are somewhat dense. Consider reorganizing them for clarity. For example, group all the IFA results in one section (e.g., Two columns -IAA and + IAA) with all the different tested markers, and place the quantification on the other side.

-IAA +IAA
BIP
GRASP
IMC
...

-Line 252: I would indicate in the text that the impact of TBC9 on bradyzoites formation was tested but that no significant difference was observed. It is still an interesting information.

-Line 464: I would precise that RAB11a is the top detected "among the potential RABs candidates detected".

-Line 537: "tend to exhibit a plaque conferring capability" sound a bit strange for me. "tend to impact plaque formation" would fit better.

Reviewer #2 (Remarks to the Author):

In this revision, Sun et al. have addressed all major and minor comments raised from my initial review. Upon further review, please address these concerns for the revised manuscript.

1. The title still seems misleading. To call TBC9 the only essential TBC domain protein in *Toxoplasma gondii* would require investigation of all lifestages, and it also discounts their potential essentiality in vivo. Furthermore, some of these genes may prove essential based on knockdown where suppressor mutations are less likely to emerge. I recommend omitting the word "only" from the title or at least specifying *Toxoplasma gondii* "tachyzoites".
2. IPR035969 finds TBC1-10, 12-18 on ToxoDB search for genes, protein features and properties, interpro domain, organism Me49. I'm not sure how the authors are finding thousands of hits with this interpro domain as mentioned in their rebuttal.
3. Line 556: Cite Sidik et al. Cell 2016 for the fitness screen.
4. Fig. 8D - should this be relabeled to TgTBC9? Remember to carefully check each gene name since they changed during revision. For an additional example, Dataset S1 uses TBC1-6HA but should now be called TBC9-6HA.
5. Fig. 9B - Add column for rank based on fold change enrichment since the Rabs are not the top hits from coIP. E.g. TBC9 Rank 1, Rab11 rank 86.
6. In methods section for Proteomic analysis of Co-IP samples, describe how fold change was calculated. It is unclear how area is calculated or why it is used instead of total spectrum counts or total unique peptide counts for each protein identified. Provide a reference for using area as a metric. State the statistical test used to calculate P values in Dataset S1. Provide a description of how area values of zero are treated. Why are there so many more zeros in the first samples vs the second sample for area?

Reviewer #3 (Remarks to the Author):

The authors provided a substantial amount of novel experiments that clarified their findings and conclusions. Notably, they now clearly demonstrated that TBC9 is localized in the ER and is involved in ER formation or structural integrity. This ER defect seems to impact on all other studied organelles, as shown and quantified by the authors : Golgi, VAC, mitochondria and IMC formation. They carefully characterized and quantified the morphological defects not only in the parasite ER and Golgi but also in other organelles, both by IFA and electron microscopy (EM).

Importantly, they have clearly demonstrated the defect in daughter cell budding.

There is still one point that requires revision before publication.

I maintain my comment, which was not changed in the revised version of the manuscript, that the role of TBC9 in rhoptry and microneme formation/secretion can not be addressed by treating the parasites with Auxin after 13h of growth (Figure 5 G-H). Mother intact organelles formed during this first phase of growth remain present and will be able to secrete their content (Fig S5). It is likely why the authors do not observe any defect in secretion.

In line, the authors wrote « To investigate the impact of TBC9 depletion on secretion of the apical organelles - the micronemes and rhoptry bulbs, we examined the secretion from these organelles in TBC9-mAID parasites induced with IAA for different time points (0, 3, and 6 hours), as described in our previous study (23). After stimulating the extracellular parasites with A23187 for 10 minutes, Western blot analysis showed no significant changes in the secretion of MIC2 into the supernatant (Figure S5A)..... »

IAA treatment of 6 hours will not impact on mother secretory organelles and allows one replication cycle at best, so very little difference between treated and not treated conditions. Why not treating for 16h in this experiment?

Thus the conclusion that TBC9 is not involved in ROP or MIC secretion should be removed from the paper as it can not be addressed by the performed experiments.

In line, figure 5 includes images always presenting 2 parasite containing vacuoles for all the studied organelle markers, those arise from this first 13h growth phase without Auxin. The fact that no additional replication cycles occur confirms that the strong defect in ER structure lead to impaired daughter cell budding correlated with impaired rhoptry/microneme formation in the daughter cells (observed by EM).

Please clarify this conclusion.

Minor corrections :

-Line 174 : « Co-localization analysis with the cis-Golgi marker GRASP (44) and the trans-Golgi network (the TGN) marker STX6 (45) showed that TBC2, 3, and 18 were closer to the trans-Golgi than the cis-Golgi (Figure 2C) »

Please replace « showed that TBC2, 3, and 18 were closer to» by « showed that TBC2, 3, and 18 display higher co-localization with the TGN than the cis-Golgi... » or « showed that the TBC2, 3, and 18 signal was more closely associated with the TGN than the cis-Golgi».

-Line 268 : « Our quantitative analysis of the Golgi at various stages showed an increased tendency towards... »

RESPONSE

Reviewers' comments:

Reviewer #1 (Remarks to the Author):

First, I would like to thank the author for addressing my concerns and responding to all my comments. A significant improvement has been made to the paper, making it easier to follow and more convincing. However, there are still some minor points that need to be addressed:

Response: We greatly appreciate the positive comments from the reviewer. We have a point-by-point response to the detailed comments and suggestions.

Minor points:

- Figure 2B: Please specify "Pearson's correlation with BIP" instead of just "Pearson's correlation" for easier understanding of the graphic.

Response: Thank you for the suggestion. We have revised the figure legend of figure 2B to "Statistics of Pearson correlation coefficient (PCC). The PCC values over the merged fluorescent region of TBC1, 5, 6, 9 and 14 with BIP were analyzed using..."

- Line 180 / Figure 2: The authors mention the potential involvement of the TBC in vesicular transport frequently. For the cytoplasmic TBC, were any colocalization studies with the ELC performed using markers like pro-M2AP?

Response: Thank you for your valuable comment. We appreciate your suggestion regarding the colocalization studies between the cytoplasmic TBC and the ELC, specifically using markers like pro-M2AP. Unfortunately, in our current study, we did not perform colocalization studies with pro-M2AP or other markers for the cytoplasmic TBC. However, we acknowledge the importance of this investigation and its potential significance for understanding the mechanism of vesicular transport involving the TBC. We will consider incorporating these experiments in future studies to further elucidate the relationship between the cytoplasmic TBC and the ELC.

- Line 423 / Figure 8: While the authors clearly demonstrate that complementation with TBC1 wt has no effect on the parasites and different markers, and the effects of the mutant TBC1 R74A and TBC1 Q101A are also clearly demonstrated in the replication and plaque assays, there is no information on the impact of mutations on the organelles previously shown to be strongly affected by TBC1 depletion (ER, Golgi, and IMC1). Are these organelles impacted in the same way?

Response: Thank you for raising this important point. We have performed additional experiments (Figure S6) that demonstrate similar mislocalization of ER, Golgi, and IMC1 in parasites complemented with TBC9 mutants upon TBC9 depletion.

- Figure 9D: The Western blot is somewhat confusing. Is the Input truly the input, corresponding to the protein before loading on the column? Does Bound correspond to

elution? Was the wash checked to ensure no more unbound TBC9 could be detected? If the input is indeed the input, how does such a small amount of Mbp-Rab11B Q71L lead to such a strong signal in the bound fraction in the last well with His-TBC9 R74A?

Response: Thank you for your questions and concerns regarding the Western blot results. We apologize for using poorly sensitive mbp antibodies in the previous experiment, which resulted in overexposed and faint bands in the Western blot results. The input and elution were matched one-to-one, and we also examined the wash buffer during the experiment to ensure no unbound TBC9 was detected. To rectify this, we have repeated the experiment and obtained new Western blot results in order to provide clear and unambiguous information (Figure 9D).

- Line 720: Some typos or indications of corrections are still present in the text

Response: Thank you for bringing this to our attention. We apologize for any reference citation errors that may still be present in the text. We have reinserted the references to ensure that the document maintains the correct citation format after formatting conversion. Thank you again for your patience and understanding in this matter.

Suggestions:

In this section I will provide some remarks for the authors to consider. The authors are free to follow them or not, as everybody as their own presentation and writing preference. These suggestions do not significantly impact the paper but could enhance understanding:

- The term "essential" is used frequently, but it should be clarified that the gene is "essential" only for the lytic cycle. As mentioned in the rebuttal letter, TBC9 KD does not affect bradyzoite differentiation and is therefore non-essential for this process and perhaps for the cat's life cycle. It should be specified that the essential aspect pertains to the lytic cycle

Response: Thank you for your comment. We appreciate your observation regarding the usage of the term "essential" in our manuscript. One point that needs clarification is that our current data on the impact of TBC9 KD on bradyzoite differentiation were obtained by inducing knockdown at 0 or 16 hours post-invasion. These data indicate that TBC9 KD rapidly leads to the rapid death or growth arrest of the parasites, making it challenging to observe its effect on bradyzoite differentiation. Further investigations require improvements in the induction conditions, such as inducing knockdown at 36 or 48 hours post-invasion, to thoroughly assess the impact of TBC9 KD on bradyzoite differentiation. Therefore, we cannot definitively state that TBC9 KD does not affect bradyzoite differentiation. We agree that it is necessary to mention that the "essential" aspect pertains to the lytic cycle. Hence, we have revised the sentence in line 250 to: "Taken together, further experiments in the type II strain ME49 supported that TBC9 is essential in the *T. gondii* lytic cycle."

- Line 37: "Facultative life cycle" sounds somewhat odd. Consider using "asexual" or "lytic cycle" instead.

Response: Thank you for your suggestion. "Facultative life cycle" is often used in biology to describe organisms that can alternate between sexual and asexual reproduction. Therefore, for *Toxoplasma gondii*, which is capable of asexual reproduction in intermediate hosts and sexual reproduction in cats, the term "facultative life cycle" may indeed be more appropriate.

- Figure 2: Removing all the "6-TY" indications in the figure labeling might improve readability.

Response: We appreciate your suggestion on the figure labeling. Removing the "6-TY" indications in the figure labeling indeed enhance readability and improve the overall clarity of the figure. We have removed all the "6-TY" in the revised version of the figure 2, as suggested.

- Figures 5/6: These figures are somewhat dense. Consider reorganizing them for clarity. For example, group all the IFA results in one section (e.g., Two columns -IAA and + IAA) with all the different tested markers, and place the quantification on the other side.

-IAA +IAA

BIP

GRASP

IMC

...

Response: Thank you for your suggestion. In fact, merging the IFA figures of Figure 5 and Figure S5 with quantitative data does make the figure appear dense. In our second version submission, we attempted to place the IFA figures on one side and the quantitative data on the other side. However, due to space limitations, we were unable to accommodate all the figures and data within Figure 5 and Figure 6. To address this issue, a possible solution is to include some of the IFA figures as supplementary material, such as the original Figure S5 (Figures 5G-I) and Figures 6B, D, and F. If you also believe this approach would help in presenting the data more clearly, we will make the necessary modifications in the next version accordingly. Alternatively, we can proceed with the current version as per your preference.

-Line 252: I would indicate in the text that the impact of TBC9 on bradyzoites formation was tested but that no significant difference was observed. It is still an interesting information.

Response: Thank you for your suggestion. As mentioned before, there is controversy regarding the impact of TBC9 knockdown on bradyzoite formation due to the rapid death or growth arrest of the parasites. Therefore, we prefer to avoid including the description of this result to prevent conveying controversial information.

-Line 464: I would precise that RAB11a is the top detected "among the potential RABs candidates detected".

Response: Great thanks to the suggestion. We have revised the sentence to "Among

the potential Rabs candidates detected, Rab11A was the top..."

-Line 537: "tend to exhibit a plaque conferring capability "sound a bit strange for me. "tend to impact plaque formation" would fit better.

Response: Great thanks to the suggestion. We have revised the sentence to "...TBC proteins tend to impact plaque formation,..."

Reviewer #2 (Remarks to the Author):

In this revision, Sun et al. have addressed all major and minor comments raised from my initial review. Upon further review, please address these concerns for the revised manuscript.

Response: We greatly appreciate the positive comments from the reviewer. We have a point-by-point response to the detailed comments and suggestions.

1. The title still seems misleading. To call TBC9 the only essential TBC domain protein in *Toxoplasma gondii* would require investigation of all lifestages, and it also discounts their potential essentiality in vivo. Furthermore, some of these genes may prove essential based on knockdown where suppressor mutations are less likely to emerge. I recommend omitting the word "only" from the title or at least specifying *Toxoplasma gondii* "tachyzoites".

Response: Thank you for the suggestion. We have revised the title to "The only essential TBC-domain protein TBC9 regulates early vesicular transport pathway and IMC formation in *Toxoplasma gondii* tachyzoites" .

2. IPR035969 finds TBC1-10, 12-18 on ToxoDB search for genes, protein features and properties, interpro domain, organism Me49. I'm not sure how the authors are finding thousands of hits with this interpro domain as mentioned in their rebuttal.

Response: Thank you for bringing up this suggestion again regarding the issue. We sincerely apologize for the misunderstanding in our previous response. We have revised the sentence as follows: "We identified 17 TBC proteins (TBC1-10, 12-18) in the genome of *T. gondii* by searching ToxoDB using the InterPro ID IPR035969 Rab-GAP-TBC domain superfamily."

3. Line 556: Cite Sidik et al. Cell 2016 for the fitness screen.

Response: Thank you for your suggestion. We have inserted the reference at the end of this sentence.

4. Fig. 8D - should this be relabeled to TgTBC9? Remember to carefully check each gene name since they changed during revision. For an additional example, Dataset S1 uses TBC1-6HA but should now be called TBC9-6HA.

Response: Thank you for pointing out this important detail. We have made the changes to label TBC1 as TBC9 in Figure 8D and Dataset S1. Furthermore, we have thoroughly reviewed the manuscript to ensure that all gene names are accurately updated in the

revised version.

5. Fig. 9B - Add column for rank based on fold change enrichment since the Rabs are not the top hits from coIP. E.g. TBC9 Rank 1, Rab11 rank 86.

Response: Thank you for your suggestion. We have incorporated a new column in Fig. 9B to indicate the rank based on fold change enrichment.

6. In methods section for Proteomic analysis of Co-IP samples, describe how fold change was calculated.

Response: Thank you for your suggestion. We have described how the fold change was calculated in the section on proteomic analysis of Co-IP samples, as suggested.

It is unclear how area is calculated or why it is used instead of total spectrum counts or total unique peptide counts for each protein identified. Provide a reference for using area as a metric.

Response: Thank you for your suggestion. The "area" refers to the quantification of peptide or protein abundance based on the integrated area under the curve of the mass spectrometry peaks. Using area instead of total spectral counts or total unique peptide counts in proteomic analysis offers the advantages of reducing errors, improving dynamic range, and simplifying data analysis. We have cited reference in the section on proteomic analysis of Co-IP samples, as suggested.

State the statistical test used to calculate P values in Dataset S1.

Response: Thank you for your suggestion. We have stated the statistical test used to calculate P values in Dataset S1.

Provide a description of how area values of zero are treated. Why are there so many more zeros in the first samples vs the second sample for area?

Response: Thank you for your suggestion. Apart from taking the average of the two replicate area values (including the zero values) to calculate fold change, we did not perform additional statistical processing on the zero values.

Mass spectrometry tends to preferentially select peptide ions from higher abundance proteins for fragmentation scanning. For lower abundance peptide ions, the fragmentation by mass spectrometry has a certain degree of randomness, resulting in these ions not being detected in every mass spectrometry analysis. Therefore, the generation of zero values primarily occurs because the protein is of extremely low abundance in the sample and was not scanned, while it was scanned and generated quantifiable values in other samples.

The number of zero values in the four samples is 244, 10, 16, and 152, with a higher occurrence in RH-1 and TBC9-2. This disparity may be attributed to inevitable differences in the sample preparation process. Additionally, variations in protein expression levels and differences in the abundance range of proteins in the samples may also contribute to the

distinct distribution of zero values.

Reviewer #3 (Remarks to the Author):

The authors provided a substantial amount of novel experiments that clarified their findings and conclusions. Notably, they now clearly demonstrated that TBC9 is localized in the ER and is involved in ER formation or structural integrity. This ER defect seems to impact on all other studied organelles, as shown and quantified by the authors : Golgi, VAC, mitochondria and IMC formation. They carefully characterized and quantified the morphological defects not only in the parasite ER and Golgi but also in other organelles, both by IFA and electron microscopy (EM).

Importantly, they have clearly demonstrated the defect in daughter cell budding.

Response: We greatly appreciate the positive comments from the reviewer. We have a point-by-point response to the detailed comments and suggestions.

There is still one point that requires revision before publication.

I maintain my comment, which was not changed in the revised version of the manuscript, that the role of TCB9 in rhoptry and microneme formation/secretion can not be addressed by treating the parasites with Auxin after 13h of growth (Figure 5 G-H). Mother intact organelles formed during this first phase of growth remain present and will be able to secrete their content (Fig S5). It is likely why the authors do not observe any defect in secretion.

In line, the authors wrote « To investigate the impact of TBC9 depletion on secretion of the apical organelles - the micronemes and rhoptry bulbs, we examined the secretion from these organelles in TBC9-mAID parasites induced with IAA for different time points (0, 3, and 6 hours), as described in our previous study (23). After stimulating the extracellular parasites with A23187 for 10 minutes, Western blot analysis showed no significant changes in the secretion of MIC2 into the supernatant (Figure S5A)..... »

IAA treatment of 6 hours will not impact on mother secretory organelles and allows one replication cycle at best, so very little difference between treated and not treated conditions. Why not treating for 16h in this experiment?

Thus the conclusion that TBC9 is not involved in ROP or MIC secretion should be removed from the paper as it can not be addressed by the performed experiments.

Response: Thank you for your suggestion. We have removed the conclusion from the paper and we set the induction time to 6 hours because depletion of TBC9 may lead to premature death or growth arrest of the parasites, making the impact on microneme and rhoptry secretion a secondary effect. Extended treatment durations could indeed provide valuable insights into the longer-term effects on parasite organelles and replication cycles,

and we will consider this for future investigations.

In line, figure 5 includes images always presenting 2 parasite containing vacuoles for all the studied organelle markers, those arise from this first 13h growth phase without Auxin. The fact that no additional replication cycles occur confirms that the strong defect in ER structure lead to impaired daughter cell budding correlated with impaired rhoptry/microneme formation in the daughter cells (observed by EM).

Please clarify this conclusion.

Response: Thank you for your suggestion. We have provided a description of the impaired formation of rhoptries and micronemes in the “Ultrastructural analysis of TBC9 depleted parasites” section, and clarified this conclusion in the discussion section: “Moreover, it was observed that the parasites did not undergo additional replication cycles (vacuoles always containing two parasites) under this induction condition. This observation provides evidence that the severe defect in the ER structure and the dispersal of the Golgi led to impaired daughter cell budding correlated with impaired rhoptry/microneme formation in the daughter cells.”

Minor corrections :

-Line 174 : « Co-localization analysis with the cis-Golgi marker GRASP (44) and the trans-Golgi network (the TGN) marker STX6 (45) showed that TBC2, 3, and 18 were closer to the trans-Golgi than the cis-Golgi (Figure 2C) »

Please replace « showed that TBC2, 3, and 18 were closer to» by « showed that TBC2, 3, and 18 display higher co-localization with the TGN than the cis-Golgi.... » or « showed that the TBC2, 3, and 18 signal was more closely associated with the TGN than the cis-Golgi».

Response: Thank you for your suggestion. We have revised the sentence as follows: “...showed that TBC2, 3, and 18 signal was more closely associated with the TGN than the cis-Golgi.”

-Line 268 : « Our quantitative analysis of the Golgi at various stages showed an increased tendency towards... »

Response: Thank you for pointing out this sentence. We have revised the sentence as follows: “Our quantitative analysis of the Golgi in different growth stages of parasite showed...”

REVIEWERS' COMMENTS:

Reviewer #1 (Remarks to the Author):

I thank the authors for their reply and all the corrections they have made since the beginning of the revision process. The article has greatly improved, and I don't have any further comments.

Reviewer #2 (Remarks to the Author):

Sun et al. have satisfactorily addressed all concerns raised by this reviewer in the second revision of the manuscript.

Reviewer #3 (Remarks to the Author):

The authors have adressed all my concerns and the revised manuscript is suitable for publication.

RESPONSE

REVIEWERS' COMMENTS:

Reviewer #1 (Remarks to the Author):

I thank the authors for their reply and all the corrections they have made since the beginning of the revision process. The article has greatly improved, and I don't have any further comments.

Response: Thank you for your kind words and acknowledgment of our efforts. We're glad to hear that the article has greatly improved to your satisfaction. We appreciate your support!

Reviewer #2 (Remarks to the Author):

Sun et al. have satisfactorily addressed all concerns raised by this reviewer in the second revision of the manuscript.

Response: We are pleased to hear that the concerns have been satisfactorily addressed. Thank you for your diligence and attention to detail throughout the review process. Thank you for your support!

Reviewer #3 (Remarks to the Author):

The authors have addressed all my concerns and the revised manuscript is suitable for publication.

Response: We're grateful for your positive feedback and glad to hear that all your concerns have been addressed in the revised manuscript. Your thorough review has been instrumental in enhancing the quality of our work. Your input is invaluable to us.